# Field observations of soil hydrological flow path evolution over 10 Millennia

Anne Hartmann[1], Ekaterina Semenova[2], Markus Weiler[2], and Theresa Blume[1]

[1]GFZ German Research Centre for Geosciences, Section Hydrology, Potsdam, Germany
[2]University of Freiburg, Chair of Hydrology, Freiburg Germany

**Correspondence:** Anne Hartmann (aha@gfz-potsdam.de)

**Abstract.** Preferential flow strongly controls water flow and transport in soils. It is ubiquitous, but difficult to characterize and predict. This study addresses the occurrence and the evolution of preferential flow during the evolution of landscapes, and here specifically during the evolution of hillslopes. We targeted a chronosequence of glacial moraines in the Swiss Alps to investigate how water flow paths evolve along with the soil forming processes. Dye tracer irrigation experiments with Brilliant Blue solution ($4\ \mathrm{g\ l^{-1}}$) were conducted on four moraines of different ages (30, 160, 3 000, and 10 000 yrs). At each moraine, three dye tracer experiments were conducted on plots of 1.5 x 1.0 m. The three plots at each moraine were characterized by different vegetation complexities (low, medium, high). Each plot was further divided into three equal subplots for the application of three different irrigation amounts (20, 40, 60 mm) with an average irrigation intensity of $20\ \mathrm{mm\ h^{-1}}$. The day after the experiment five vertical soil sections were excavated and the stained flow paths were photographed. Digital image analysis was used to derive average infiltration depths and flow path characteristics such as the volume and surface density of the dye patterns. Based on the volume density, the observed dye patterns were assigned to specific flow type categories. The results show a significant change in type of preferential flow paths along the chronosequence. The flow types change from a rather homogeneous matrix flow in coarse material with high conductivities and a sparse vegetation cover at the youngest moraine to a heterogeneous infiltration pattern at the medium-age moraines. Heterogeneous matrix and finger flow are dominant at these intermediate age classes. At the oldest moraine only macro pore flow via root channels was observed in deeper parts of the soil, in combination with a very high water storage capacity of the organic top layer and low hydraulic conductivity of the deeper soil. In general, we found an increase in water storage with increasing age of the moraines, based on our observations of the reduction in infiltration depth as well as laboratory measurements of porosity. Preferential flow is, however, not only caused by macropores, but especially for the medium age moraine seems to be mainly initiated by soil surface characteristics (vegetation patches and micro-topography).

*Copyright statement.* TEXT

# 1 Introduction

The ability of soil to store and to transport water is essential for its ecosystem services such as nutrient cycling or water and gas balances [Clothier et al. (2008), Amundson et al. (2015), L. Hatfield et al. (2017), Shang et al. (2018)]. Thus, the interaction of water and soil is an elementary foundation for the existence and functioning of terrestrial ecosystems. This interaction is part of a large network of interactions of various ecosystem components (flora, fauna, material and energy fluxes, geomorphological conditions, climate), which are also necessary for the existence and functioning of ecosystems. Soil filters the percolating water, redistributes it to groundwater or stream water or holds it against gravity and makes it available for plants.

The soil functions are influenced and controlled by soil properties, which can vary spatially on the small (Hu et al., 2008) and large scale (vertically along the profile and horizontally across landscapes (Bevington et al., 2016)), as well as temporally. These properties include soil texture and structure, i.e. the pore- and grain size distribution which in turn control the storage- and transport capacity of the soil. Additional factors influencing soil functions are climate, topography and vegetation. In undisturbed natural systems these factors are usually assumed to be constant at the observational time scale and the inherent system dynamics only become apparent on long time scales.

Preferential flow, which is defined according to Hendrickx and Flury (2001) as a phenomenon 'where water and solutes move along certain pathways, while bypassing a fraction of the porous matrix', has impacts on water storage (Rye and Smettem, 2017) and thus plant water availability. It furthermore affects the transport of nutrients and contaminants (Jarvis, 2007) throughout the vadose zone and consequently also soil chemistry [Jin and Brantley (2011), Bundt et al. (2000)] and groundwater quality. Allaire et al. (2009) attributes rapid flow and mass transport to flow through earthworm burrows, cracks in soil, and to flow paths resulting from soil layering and hydrophobicity. They defined four types of preferential flow: crack flow, burrow flow (created by soil fauna), finger flow, and lateral flow along layer interfaces, where flow in burrows and cracks is also often classified as macropore flow. We will in the following distinguish flow in macropores according to their origin as crack flow and biopore flow, where the latter includes channels by activities of roots and soil fauna. Preferential flow in form of macropore flow occurs mostly in fine-textured soils whereas finger and funnel flow rather occurs in soils with a coarse texture (Hendrickx and Flury, 2001). General factors which can cause preferential flow paths are surface structure and properties such as vegetation cover, micro topography or hydrophobicity, as well as subsurface soil properties such as soil structure and soil type, subsurface heterogeneities, flow instabilities and plant root activities [Weiler and Naef (2003), Clothier et al. (2008), Bachmair et al. (2009) Jarvis (2007), van Schaik (2009), Wang et al. (2018)]. Soil water conditions were also found to have an influence on the preferential flow path characteristics [Gimbel et al. (2016), Hardie et al. (2011), Bogner et al. (2008)].

Many preferential flow-influencing properties such as soil structure, soil texture or vegetation cover change during landscape evolution [e.g. Vilmundardóttir et al. (2014), Egli et al. (2010), Dümig et al. (2011)] and thus also lead to a change in the soil hydraulic behavior (Lohse and Dietrich, 2005) which in turn has a direct impact on the surface and subsurface water transport. We therefore assume that the age of the soil has an influence on the prevailing preferential flow paths, and thus the type and also the depth extent of the preferential flow paths can change over time. Especially root activities can lead to generation of preferential flow paths in deeper layers, which was found by Cheng et al. (2014) based on a comparison of young and older

forest plantations. On a large time scale of several million years Lohse and Dietrich (2005) found a transition from mainly vertical water transport in younger volcanic soils in the Hawaiian Islands to lateral water transport along the boundary of a subsurface clay layer. The younger soil was coarse textured with high saturated hydraulic conductivities along the profile and a rather low field capacity, whereas the older soil revealed a higher field capacity and a distinct reduction in saturated hydraulic

conductivity throughout the profile due to clay accumulation. Yoshida and Troch (2016) observed a major change in flow paths from deep groundwater flow to shallow subsurface flow in volcanic catchments of ages between 200 000 and 82 million years. While the change of major flow paths with time has already been studied at the time scale on the order of 100 000 to millions of years, little is known how flow paths change during these first 10 000 years of landscape development.

Areas with receding glaciers have been shown to be suitable for soil development studies [ Crocker and Dickson (1957),

Douglass and Bockheim (2006), He and Tang (2008), Egli et al. (2010), Dümig et al. (2011), Vilmundardóttir et al. (2014), D'Amico et al. (2014)]. In the cool and humid climate regions of former glacial areas the soils develop from mineral soils to soils with a highly organic topsoil. These organic soil types are less intensively studied with regard to their soil hydraulic behavior compared to mineral soils (Carey et al., 2007). It is known that these soils differ in their soil hydraulic properties from mineral soils (high total porosity (up to 90%) and a low bulk density (Carey et al., 2007)) but little is known about how this

development impacts water flow paths. Therefore, this study addresses the occurrence and the evolution of preferential flow during the first 10 000 years of landscape evolution in glacial moraines in the Swiss Alps. More specifically, we test the hypotheses that (1) Vertical subsurface flow path types and vertical extent of flow paths change through the millennia as: (2) The proportion of macropore flow will increase due to the development of biopores, (3) The soil develops from a homogeneously mixed material into a depth differentiated soil system, and (4) Physical weathering leads to a reduction in particle size and an

increase in porosity.

Dye tracer experiments, and an analysis of soil texture and soil physical properties were used to investigate how water flow paths evolve with hillslope age. The hydropedological approach (Lin, 2003) that links pedon (Quisenberry et al., 1993), landscape (Cammeraat and Kooijman, 2009) and hydrologic processes studies has already been applied to the preferential flow phenomenon (Jarvis et al., 2012). Dye tracer experiments combined with digital image processing have been applied success-

fully to study preferential infiltration in soils [Weiler (2001), Bogner et al. (2008), Blume et al. (2008), Laine-Kaulio et al. (2015), Hardie et al. (2011), Cheng et al. (2014)]. In our study we use this method to identify how flow paths change during the co-evolution of soil, vegetation and topography. Understanding the changes in preferential flow paths as a result of the natural co-evolution of landscape forming factors can provide valuable knowledge on how these systems can also change as a result of human intervention (Richter and Mobley (2009)).

## 85   2   Material and methods

### 2.1   Study site

The study area is located in the foreland of the Stein glacier above the tree line in the Central Swiss Alps, south of the Sustenpass in the Urner Alps (appr. 47° 43'N, 8° 25'E). Elevations range from 1900-2100 m a.s.l. The area lies in the polymetamorphic

"Erstfelder" gneiss-zone, which is part of the Aar-massif (Blass et al., 2003). The geology is defined by metamorphosed pre-Mesozoic, metagranitoids, gneisses, and amphibolites [Heikkinen and Fogelberg (1980), Schimmelpfennig et al. (2014)], thus the material is mainly acidic and rich in silicate. The closest official weather station is located 18 km away at Grimsel Hospiz (46° 34'N, 8° 19'E) at an elevation of 1980 m a.s.l. For the norm period from 1981 to 2010 the station recorded an annual mean temperature of 1.9 °C and an annual precipitation of 1856 mm. The precipitation distribution throughout the year is fairly uniform with a slight increase in the winter months (Schweizerische Eidgenossenschaft, 2016). The glacier foreland consists of moraines with unconsolidated glacial till. The humid and cool climate together with the nutrient-poor substrate and a relative high water permeability of the glacial till favor the formation of podsolic soils and humus in this area (Heikkinen and Fogelberg, 1980).

The moraines of the Stein glacier were exposed due to its retreat to the south. Four moraines were selected for this study (see Fig. 1). Schimmelpfennig et al. (2014) conducted a detailed dating study of the Stein glacier moraines, based on high-sensitivity beryllium-10 moraine dating and found that the ages of three moraines range between 160 to 10 000 years. The age of a fourth moraine was dated to 30 years based on maps and aerial photos.

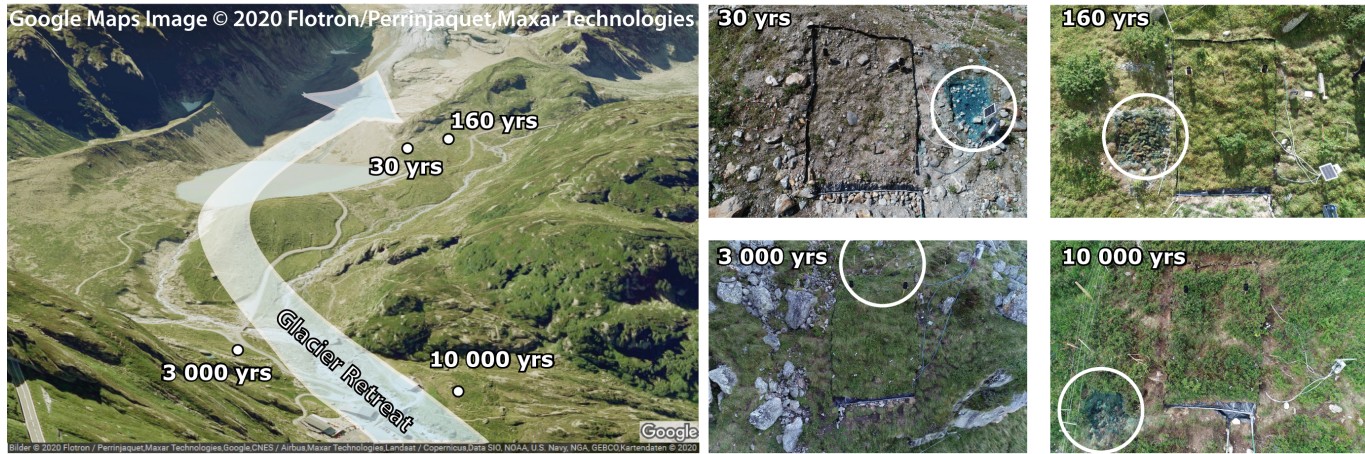

**Figure 1.** Location (left) and surface cover (right) of the four selected proglacial moraines of the Stein glacier. White circles show locations of one of the three brilliant blue experiment plots per age class. Photo of location is provided by Google (n.d.). Photos of the 30, 160, and 10 000-year-old moraine were taken after the brilliant blue experiment (photos taken by F. Lustenberger).

The oldest moraine with an age of 10 000 years is facing north-east. The second oldest moraine with an age between 2 000 and 3 000 years is the only one facing south. The two youngest moraines were exposed in the years 1860 and 1980-1990, and thus have an age of 160 and 30 years, respectively. Both moraines are facing north-east and are located closer to the glacier tongue at a distance of approximately 1 km from the oldest moraines. Both moraines are south of the glacial lake "Steinsee" (1930 m a.s.l.), which is a proglacial lake that was formed by the glacier retreat in 1924 (Blass et al., 2003).

The vegetation of the moraines was mapped in summer 2017 (Maier et al., 2019). Pronounced differences in vegetation cover-

age and species distributions were found among the four age classes. The vegetation of the oldest moraine was dominated by a variety of prostrate shrubs together with small trees and several grasses. On the 3 000-year-old moraine a grassland cover with fern, mosses, sedges and forbs was found. The two youngest moraines, however showed a lower degree of vegetation complexity. On the 160-year-old moraine a combination of grasses, lichen, forbs, and shrubs was present. The youngest moraine still shows only a sparse vegetation cover with mainly grass, moss, forbs, and a few shrubs.

## 2.2 Field experiments

The dye tracer experiments were conducted between mid July and mid August 2018. We used Brilliant Blue FCF as dye tracer due to its good visibility, high mobility and non-toxicity. We used a concentration of 4 g $l^{-1}$ at which the tracer shows good sorption and visibility (Weiler and Flühler, 2004). Three study plots were selected at each moraine, based on degree of vegetation complexity (low, medium and high complexity). Vegetation complexity is characterized by vegetation coverage, number of species and the plant functional diversity. The functional diversity is calculated based on specific leaf area, nitrogen content, leaf dry matter content, Raunkiaérs life form, seed mass, clonal growth organ, root type and growth form. The collection of the required data and calculation of the vegetation complexity was done by the Geobotany Group of the University of Freiburg and is described in more detail in Maier et al. (2019).

The size of each study plot was 1.5 x 1.0 m. The distances between the three study plots at each moraine ranged from 10 to 100 m. Each plot was further divided into three equal subplots of 0.5 x 1.0 m. Figure 2 shows the experimental design at each moraine and illustrates the irrigation procedure.

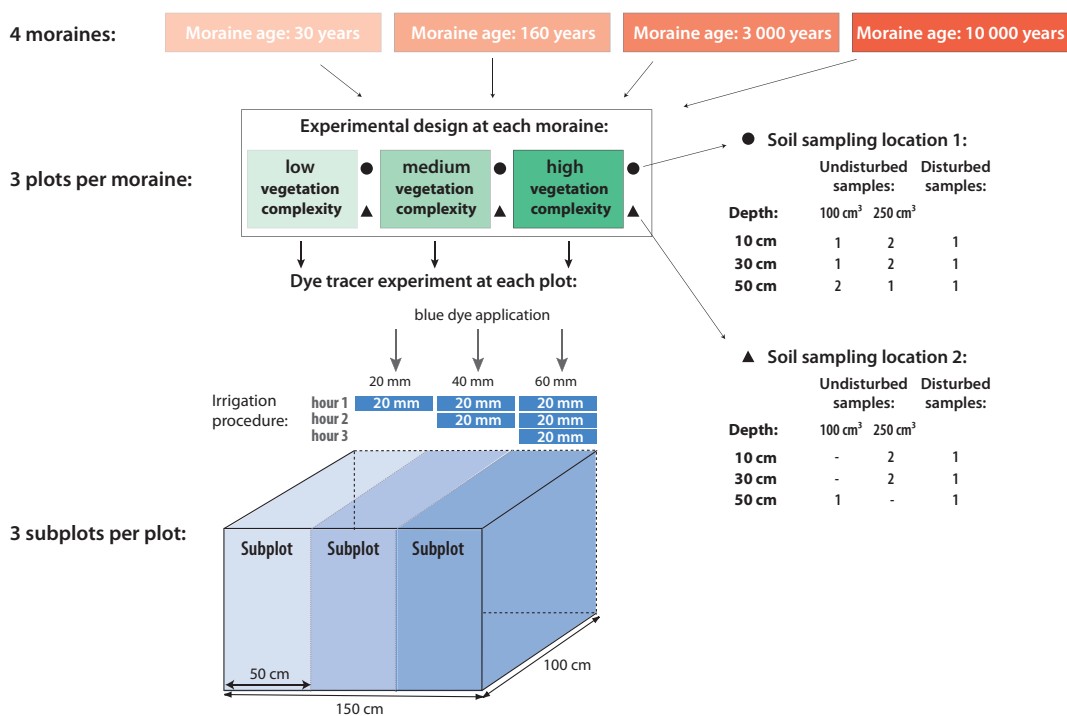

**Figure 2.** Illustration of the experimental design and soil sampling scheme at each moraine.

The subplots were irrigated with three different amounts of dyed water (20 mm, 40 mm, 60 mm) and an irrigation intensity of 20 mm h$^{-1}$. The irrigation intensity is relatively high with a return period of 2.8 years (Fukutome et al., 2017). In preparation of the tracer application large vegetation in form of shrubs and bushes were cut off to a height of a few centimeters to reduce interception. The tracer was applied with a hand-operated sprayer connected to a battery powered pump which guaranteed a constant pressure for a uniform flow rate of 60 l h$^{-1}$. For a time-efficient irrigation of the three subplots with three irrigation

amounts, the irrigation procedure was divided into three steps. In the first step all three subplots were irrigated simultaneously for 60 min in a sequence of 5 min irrigation and 5 min break. This provides an application of 20 mm to all three subplots. After finishing the first step the first subplot was covered to avoid any additional water input. In a second step, the other two subplots were simultaneously irrigated for additional 60 min in a sequence of 5 min irrigation and 10 min breaks. This provides an application of additional 20 mm to each of the two remaining subplots. In the last step only the third subplot was irrigated for

60 min in a sequence of 2 min irrigation and 10 min breaks while the other two plots remained covered providing an additional 20 mm to this subplot. After the end of tracer application, the entire plot was covered to avoid any disturbance by natural rainfall.

The next day each subplot was excavated in up to five profiles of 7 to 10 centimeters. After the profile cuts were made with pickaxes, spades and hand shovels, the profile walls were cleaned. Hanging roots were cut off and rocks were not removed but

made visible. The profiles of each subplot were photographed with a Panasonic Lumix DMC-FZ18 camera and a resolution

of 2248 x 3264 pixels. A big umbrella was used to provide a uniform light distribution in the photographs and to avoid direct sunlight. A wooden frame for a geometric correction and a gray-scale (Kodak) attached to the frame (Fig. 3) for a later color adjustment were included in the photographs. Since dye tracer experiments only provide snapshots of flow patterns at 24 h after the irrigation, we cannot exclude the possibility that initial preferential flow paths were obliterated by a later downward movement of the infiltration front. However, as the probability for this special case is relatively low, we assume that these snapshots are a viable basis for the comparison of characteristic flow patterns along the moraine ages.

### 2.3 Image analysis

The image analysis procedure by Weiler (2001) was used to generate tri-color images of the photographs showing stained and unstained areas (Fig. 3). A detailed description of the method can be found in Weiler and Flühler (2004). Instead of the original IDL software package a similar Python version was used. Basically, a geometric correction, a background subtraction and color adjustment was carried out to correct differences in image illumination and changes in the spectral composition of daylight. The delineation of rocks and plants was done manually. In the resulting tri-color image the horizontal and vertical length of a pixel correspond to 1 mm. Due to poor lighting conditions or a heterogeneous background color distribution in the soil caused by material transitions, small stones or organic matter, the image analysis software was not able to recognize all large dye stains as coherent objects. Thus, a manual correction of the images using the photographs was necessary (see Fig 3).

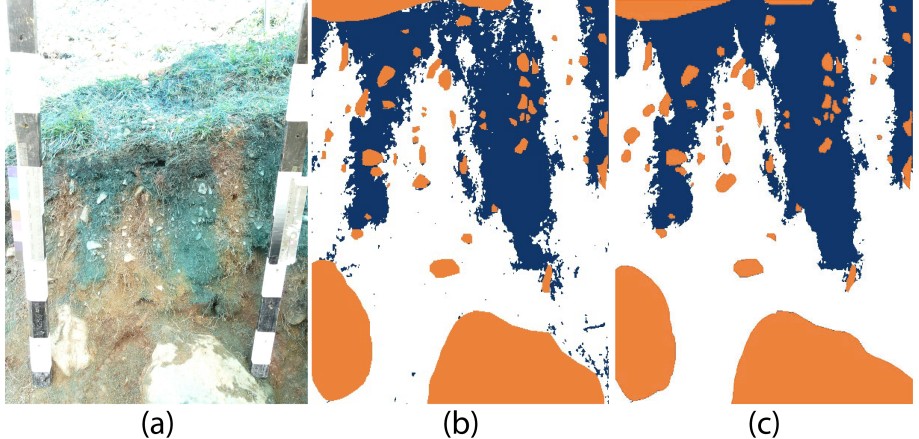

(a)  (b)  (c)

**Figure 3.** Exemplary image analysis procedure for the subplot irrigated with 40 mm at the 3 000-year-old moraine: (a) Photograph of the vertical soil profile with wooden frame and attached gray-scale. (b) Software generated tri-color image of the photograph. (c) Manually corrected tri-color image. Blue indicates stained soil, white unstained soil, and orange indicates rocks.

For a quantitative comparison of the dye patterns the maximum infiltration depth, the volume density and surface area density, as well as the stained path width were calculated. These parameters are frequently used for an objective comparison and description of the dye patterns [Weiler and Flühler (2004), Bachmair et al. (2009), Laine-Kaulio et al. (2015), Cheng et al. (2014), Gimbel et al. (2016), Laine-Kaulio et al. (2015), Mooney and Morris (2008) Öhrström et al. (2002)]. Volume and

surface area density are originally steorological parameters which are used to relate three-dimensional structures to measured two-dimensional parameters (Weibel, 1979). The volume density corresponds to the dye coverage and can be derived from one-dimensional information by calculating the fraction of stained pixels for each depth. The volume density profile is defined by the fraction of stained pixels per depth and is calculated as the average of all excavated profiles per plot. The surface area density in one-dimension is calculated by using the intercept density, which describes the number of intercepts between stained

and unstained pixels divided by the horizontal width of the soil profile. The profile of the surface area density describes the amount of intercepts per depth and is then also averaged over all photographed profiles per plot. Volume density provides no information whether the stained area is the sum of many small fragments or a few large ones, thus the volume density alone should not be used to characterize flow patterns and the surface area density should be used as a supplementary parameter. A high surface area density indicates a large number of small features.

Following the method described by Weiler (2001) the resulting dye patterns were next classified into flow type categories based on the proportions of three selected stained path width classes (stained path width <20 mm, 20 mm-200 mm, >200 mm) relative to the volume density. The stained path width is equal to the horizontal extent of a stained flow path (Weiler, 2001). This classification method distinguishes between five flow types: (1) macro pore flow with low interaction, (2) mixed macro pore flow (low and high interaction), (3) macro pore flow with high interaction, (4) heterogeneous matrix flow/finger flow, and

(5) homogeneous matrix flow. Dye patterns, which cannot be classified as one of these flow types are categorized as undefined. The classification method based on proportions of the stained path width classes is illustrated in Figure 4.

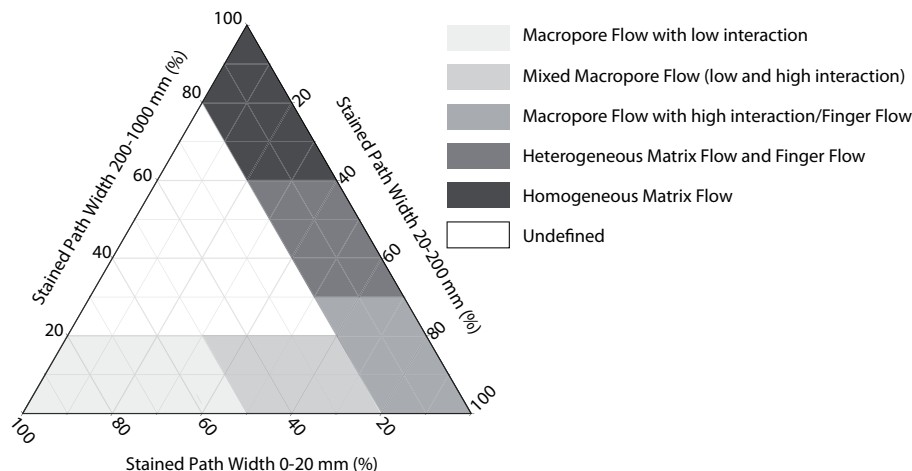

**Figure 4.** Flow type classification based on the proportion of the three stained path width classes: ternary diagram after Weiler (2001).

This method is based on the assumption that the dye patterns are mainly controlled by certain preferential flow processes and each flow process creates a characteristic dye pattern that can be described by the extent and distribution of stained features. This method was proven to be suitable for the investigation of Weiler (2001) and Weiler and Flühler (2004) and was also

used in a variety of additional studies [Bachmair et al. (2009), Gimbel et al. (2016), Mooney and Morris (2008)]. However, an

extension of the flow type categorization was needed for our study site with soils of different ages, texture and additionally a high stone content. In the extended classification, we avoid a clear differentiation between macro pore flow and finger flow. The original classification assigns finger flow only when both the medium sized stained path width class (20-200 mm) and the class with the biggest stained path widths (> 200 mm) account for approximately half of the dye coverage. This implies that finger flow is only prevalent when the dye pattern is characterized by a majority of broader stained path widths, ignoring that the size of the finger-like flow paths can vary over a broad range (Wang et al., 2018). Thus, we argue that while finger flow and macro pore flow are caused by different properties and flow mechanisms, they can lead to similar dye patterns and distributions of stained path width classes. This is especially the case for macro pore flow with high interaction, which creates broader stained paths that could also be assigned to finger flow. Therefore, both flow types were considered in the extended classification for this class. Furthermore, it was observed that the presence of rocks within the image analysis interrupts homogeneous blue stained areas and thus leads to smaller stained path widths. Using the original classification scheme on a soil profile with high stone content suggests a heterogeneous flow pattern, which can be classified as heterogeneous matrix flow, finger flow, or macro pore flow depending on the abundance of rocks. Therefore, an additional class has been introduced, which is used when homogeneous matrix flow between rocks takes place. The classification rule for the additional flow type class is based on the proportion of blue dye coverage of the available permeable matrix space (profile width minus sum of stone widths per row). If at least 95 % of the permeable space is stained by blue dye the flow type is classified as matrix flow between rocks.

## 2.4 Soil sampling and laboratory analysis

Soil samples were taken during August and September of 2018 close to each dye tracer plot. For grain size analysis, two disturbed bulk soil samples per depth were taken at 10, 30, and 50 cm depth at each plot. The total of 72 samples was analyzed in the laboratory between November 2018 and January 2019 by using a combination of dry sieving (grain sizes > 0.063 mm) and sedimentation analysis (grain sizes < 0.063 mm) with the hydrometer method (Casagrande, 1934). Organic matter removal was only possible by floating off the lighter fractions prior to grain size analysis. Since three plots were selected per moraine for the brilliant blue experiments, six samples per depth and age class (a total of 18 samples for each moraine) were available for the grain size analysis. Grain sizes between 2 mm and 0.063 mm were classified as sand, between 0.063 mm and 0.002 mm as silt and grain sizes smaller than 0.002 mm as clay. Grain size fractions of particles < 2 mm were calculated as weight percentages of total weight of particles < 2 mm, thus excluding gravel and stones to avoid that single larger stones shift or dominate the distribution. The gravel and stone fraction was calculated separately as a weight percentage of the entire soil sample.

For the analysis of the structural parameters soil samples were taken with sample rings to provide undisturbed cores which preserve the natural soil structure. At each plot two 250 cm$^3$ and one 100 cm$^3$ undisturbed soil samples were taken at a depth of 10 and 30 cm. Three samples of 100 cm$^3$ were taken at 50 cm depth. Thus, per age class nine samples per depth were available for the determination of the structural parameters porosity and bulk density. A detailed overview of the sampling scheme at each plot is given in Figure 2.

The porosity was determined in the lab using the water saturation method. For this method sample weights were recorded at

saturation and after drying at 105 °C. For saturation, the samples were placed in a small basin. The water level in the basin was increased step wise by 1 cm per day. When the water level reached the top of the soil sample and the sample was fully saturated, the bottom of the sample was sealed and the weight at saturation was measured. Bulk density was determined by relating the dry mass after drying at 105 °C to the sample volume. The loss on ignition is a measure of the organic substance in the soil and describes the proportion of the organic substance that was oxidized during annealing for 24 hours at 550 °C. The loss on ignition was determined by drying sub-samples (4-6 g) for at least 24 hours at 105 °C and then at 550 °C. The ignition loss is then calculated by relating the weight loss after drying at 550 °C to the sample weight after drying at 105 °C.

## 2.5 Statistical analysis

The non-parametric Kruskal-Wallis test was used to test the significance of the differences in the soil texture among the four moraines of differing age classes. It can be applied when the assumption of a normal distribution can not be made and is also valid for small sample sizes. We applied the test to each grain size fraction across the four age classes. Average values were based on 18 samples per age class. The grain size distribution at 10 cm depth of the oldest moraine was excluded from consideration, as due to the high organic content not all organic matter could be removed and the results may therefore be erroneous.

## 3 Results

### 3.1 Soil texture and structural parameters

Comparing the depth averaged soil texture over the millennia we find that while the soil texture at the youngest moraine mainly consists of sand, the grain sizes decrease over the millennia with silt being the largest fraction after 10 000 years (Fig. 5). Clay content increased with age for the three older moraines, with the youngest moraine being the exception (having the second highest clay content). The Kruskal-Wallis test with a 0.05 confidence level showed that differences in grain size fractions among the four age classes were statistically significant (p-values < 0.05; sand: p=0.0013, silt: p=0.0006, clay: p=0.0018).

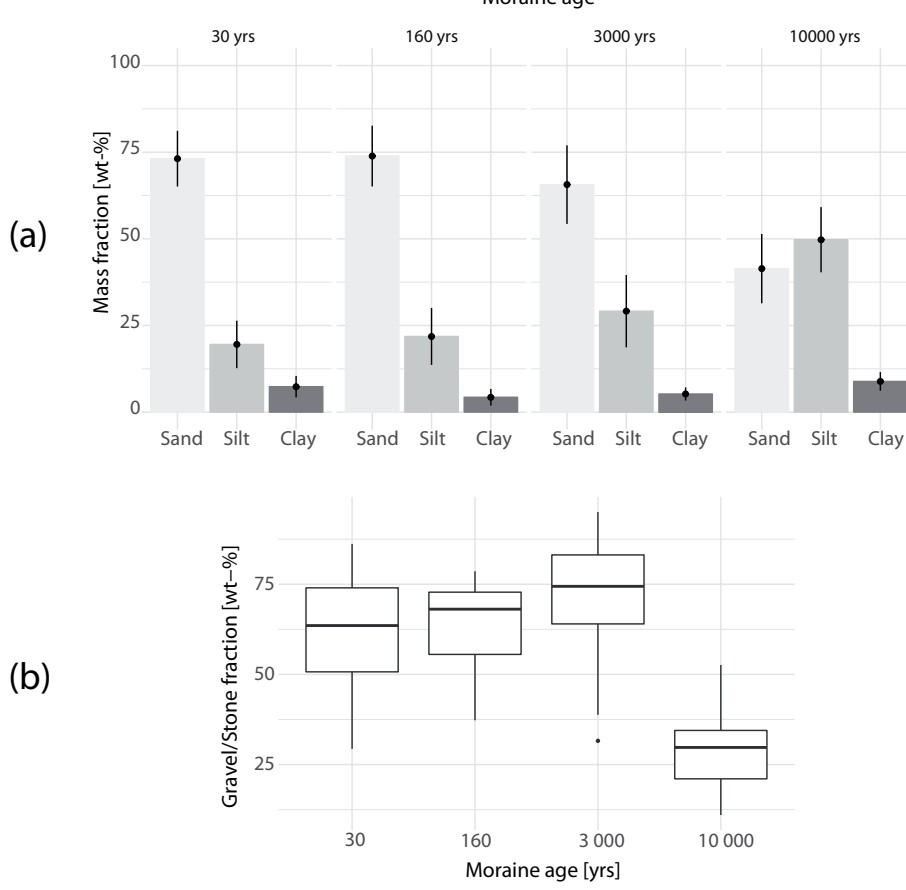

**Figure 5.** (a) Profile averaged grain size fractions for the four moraines. Fractions are percentages of the fine earth fraction (< 2 mm). (b) Profile averaged gravel content (> 2mm) calculated as the percentage of the entire sample weight. Each average is based on 18 samples.

A significant reduction in grain size over time was observed, which is most pronounced between 3 000 and 10 000 years of soil development. The gravel and stone fraction is roughly the same at the three younger moraines and significantly lower at the oldest moraine (Fig. 5(b)). The structural parameters porosity and bulk density also show a clear trend with age, with porosity increasing and bulk density decreasing (Fig. 6).

**Table 1.** Linear rates of change in porosity, bulk density and grain size between adjacent age classes calculated based on median values.

| | Porosity $[yr^{-1}]$ | | | Bulk Density $[g\ cm^{-3}yr^{-1}]$ | | | Grain Size $[mf - \%\ yr^{-1}]$ | | |
|---|---|---|---|---|---|---|---|---|---|
| | $10\ cm$ | $30\ cm$ | $50\ cm$ | $10\ cm$ | $30\ cm$ | $50\ cm$ | sand | silt | clay |
| Period $[yrs]$ | | | | | | | | | |
| $30-160$ | $5.1\text{x}10^{-4}$ | $3.2\text{x}10^{-4}$ | $1.1\text{x}10^{-4}$ | $-1.3\text{x}10^{-3}$ | $-1.4\text{x}10^{-3}$ | $-9.1\text{x}10^{-4}$ | $2.5\text{x}10^{-2}$ | $1.5\text{x}10^{-2}$ | $-2.0\text{x}10^{-2}$ |
| $160-3\,000$ | $9.5\text{x}10^{-5}$ | $4.6\text{x}10^{-5}$ | $4.4\text{x}10^{-5}$ | $-2.9\text{x}10^{-4}$ | $-9.7\text{x}10^{-5}$ | $-1.4\text{x}10^{-4}$ | $-3.2\text{x}10^{-3}$ | $3.2\text{x}10^{-3}$ | $3.1\text{x}10^{-4}$ |
| $3\,000-10\,000$ | $1.6\text{x}10^{-5}$ | $3.3\text{x}10^{-5}$ | $2.6\text{x}10^{-5}$ | $-7.6\text{x}10^{-5}$ | $-7.4\text{x}10^{-5}$ | $-4.5\text{x}10^{-5}$ | $-3.3\text{x}10^{-3}$ | $2.4\text{x}10^{-3}$ | $3.2\text{x}10^{-4}$ |

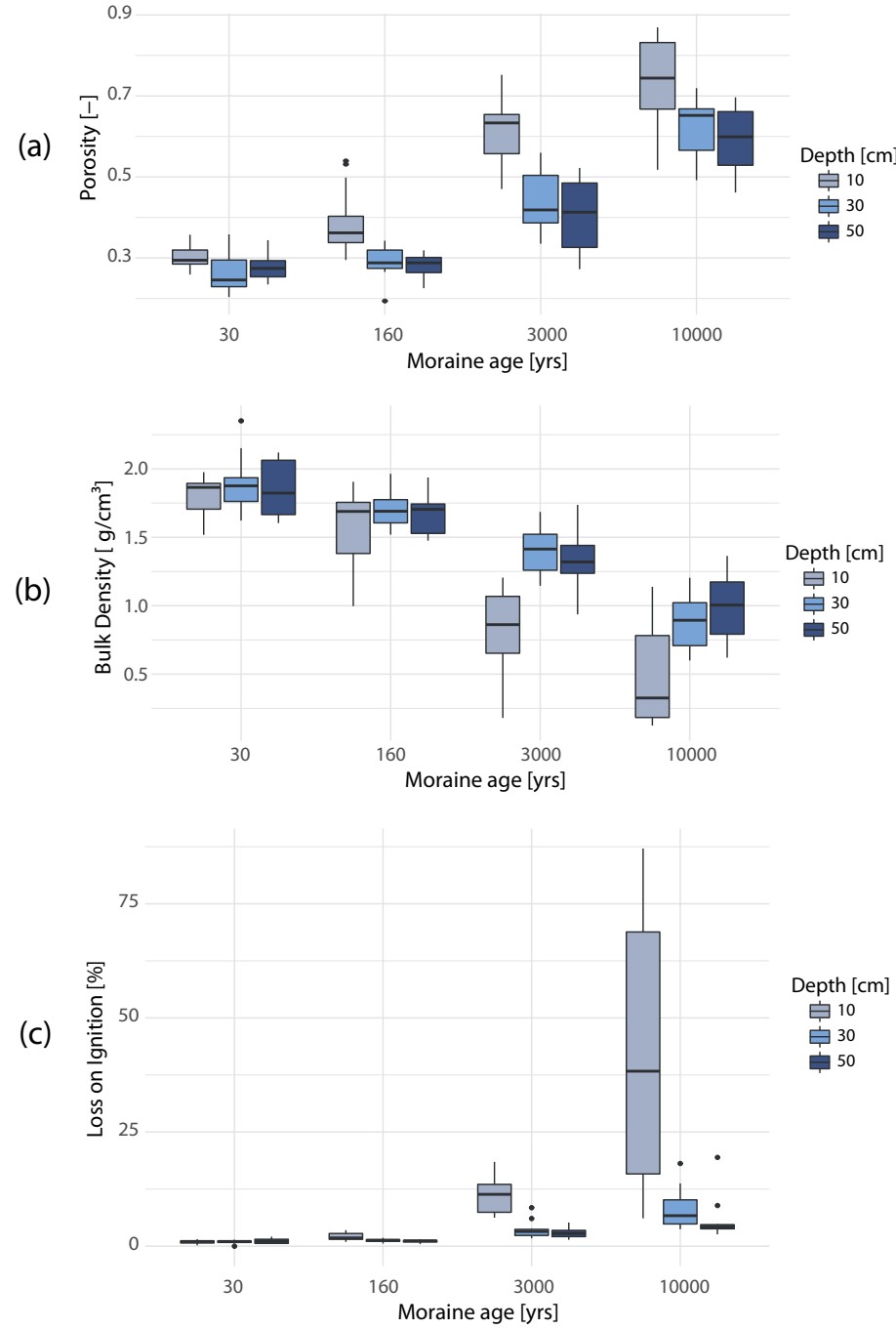

**Figure 6.** Evolution of soil porosity (a) bulk density (b) and loss on ignition (c) in 10, 30, and 50 cm depth.

The porosity observed at the youngest moraine ranges between 0.22 and 0.37, with no pronounced differences among the three soil depths. The 160-year-old moraine has a higher porosity in the upper 10 cm than the 30-year-old moraine. After 3 000 years the increase in porosity continues but is now also visible in 30 and 50 cm, but with much higher values at 10 cm (Fig. 6(a)). After 10 000 years the porosity at 10 cm ranges from 0.6 to up to more than 0.8. The other two depths also experienced a further increase in porosity. The decrease in bulk density is also most pronounced in the top layer of the soil (Fig. 6(b)). While

after 30 years the bulk density in the upper 10 cm ranges around 1.7 g cm$^{-3}$ the bulk density after 10 000 years is much smaller and ranges between 0.2 and 0.7 g cm$^{-3}$. After 3.000 years the trend is also visible in 30 and 50 cm.

The loss on ignition, as a measure for the organic matter content, shows an increase throughout the first 10 millennia of soil development, which is most pronounced in the upper soil layer (see Fig. 6(c)). At the two youngest moraines the organic matter content is still very low (< 2 weight-%). At these two age classes the organic matter content is homogeneously distributed over

the profile, with a slight tendency to higher values in the topsoil at the 160-year-old moraine. The 3 000-year-old moraine shows a strong increase in organic matter content in the surface layer. At the oldest moraine the trend of increasing organic matter continues in all three depths. Here, the organic matter content in the topsoil makes up to two-thirds of the soil material. However, the organic matter content varies distinctly with a minimum of 6 weight-% and a maximum of 87 weight-%. In deeper depths, the organic content also increases compared to the 3 000 year old soil, but remains below 20 weight-%.

Even though the differences in soil physical characteristics between 30 and 160 years are comparatively small, the rates of change during this initial phase are highest (Tab. 1). Between 160 and 3 000 years, the rates of change are significantly reduced and remain in a similar range between 3 000 and 10 000 years.

## 3.2   Vertical dye pattern analysis

Flow patterns traced with brilliant blue dye changed considerably with moraine age (Fig. 7). The volume density profile is a

measure for the amount of blue dye per depth. The profile patterns of volume and surface area density show distinct differences among age groups, while differences between the vegetation complexity levels are not as clear (Fig. 7 and Fig. 8).

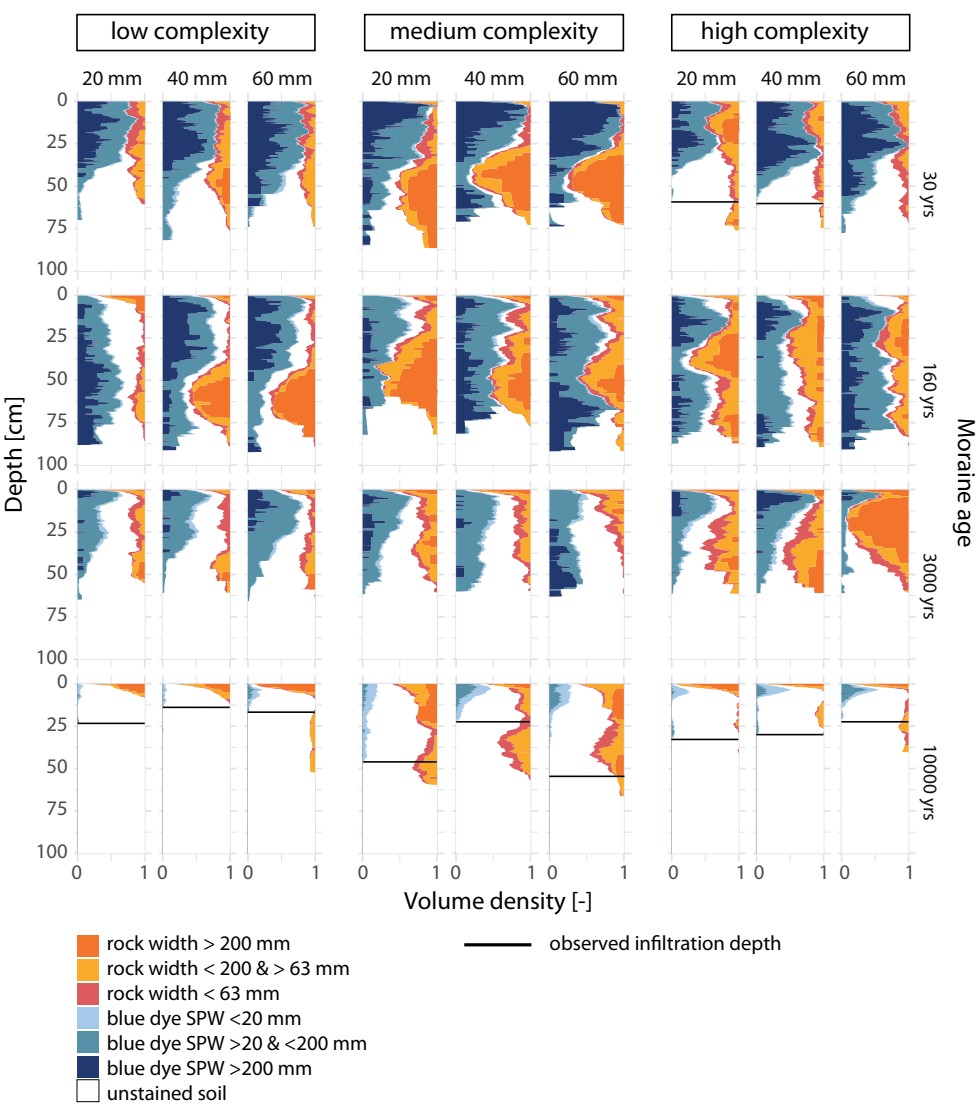

**Figure 7.** Volume density profiles per age class, vegetation complexity and irrigation amount. The volume density is the fraction of stained pixels, here colored by flow path width (stained path width, SPW) and rock sizes.

The volume density of the blue dye was classified in three selected stained path width groups (Weiler, 2001). Additionally, the volume density of rocks was also classified in three groups (Fig. 7).

An analysis of the average or maximum infiltration depth based on the dye profiles was not possible because not all profiles

could be excavated up to the maximum infiltration depth. In most cases large boulders prevented further excavation or the infiltration depth was more than one meter. The latter was mostly the case at the youngest moraines. Only at the oldest moraine a maximum infiltration depth could be determined based on the dye profiles.

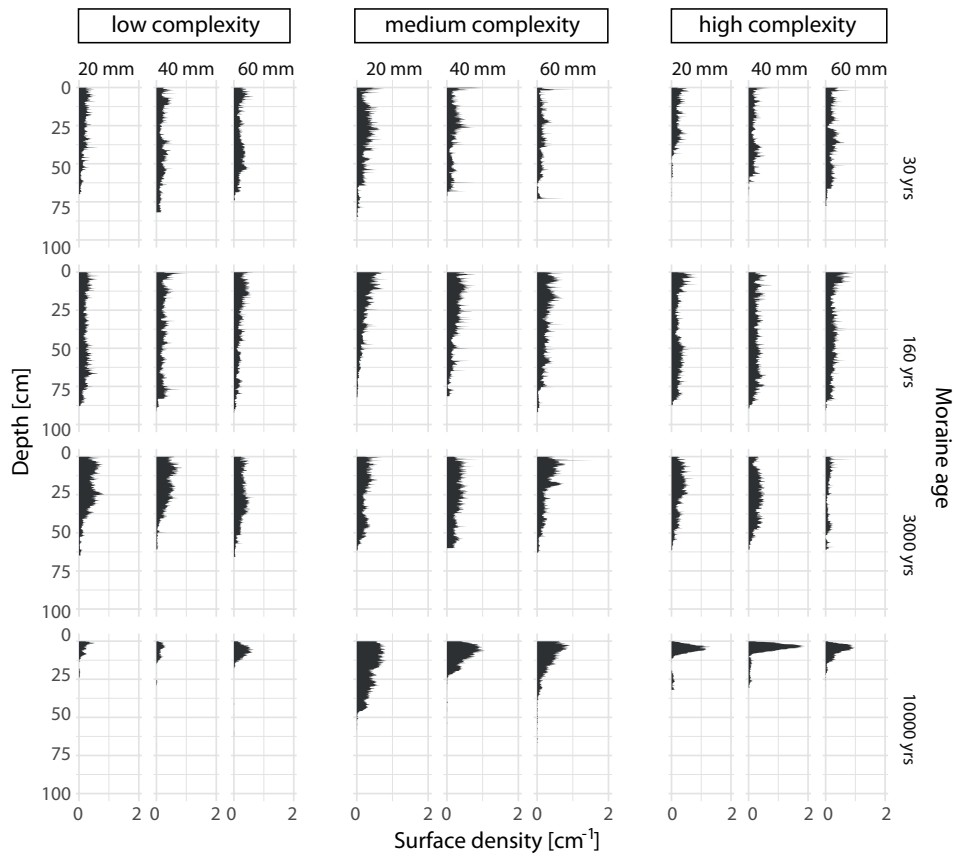

**Figure 8.** Surface area density profiles per age class, vegetation complexity and irrigation amount (20, 40 and 60 mm). A high surface area density indicates a large number of small features.

Combining the volume density profiles with the surface area density profiles, it is possible to derive whether the stained area described by the volume density is made up of many small flow paths or few large ones. The shapes of the volume density
profiles and the surface area density of the youngest moraine are all very similar across the vegetation complexity levels and irrigation amounts. The youngest moraine has a higher volume density of flow paths in the top half of the soil profile than all other moraines (Fig. 7). There are almost no unstained areas. Beginning from approximately 30 cm depth the volume density

declines. The surface area density profiles show an opposite pattern (Fig. 8). The surface area density is smaller in the upper half and increases in the lower half of the profile. The combination of both parameters indicate a homogeneous staining in the

275 top half, where interruptions of stained areas are only caused by rocks. In the lower part of the soil profiles the flow paths are subdivided, which is indicated by the increase in surface area density (apparent at 30 cm depth for the low, 25 cm depth for medium and 50 cm depth for the high vegetation complexity plot). This combined with a decline in volume density indicates a narrowing of the flow paths. For the low and high vegetation complexity plots the change in flow paths coincides with a layer of higher clay and silt content. This layer does not exist at the medium vegetation complexity plot. In this case, the narrowing

and splitting up of flow paths is caused by large rocks. No clear differences are visible between the different irrigation amounts. The proportion of the stained path width classes are controlled by the existence of rocks. The maximum infiltration depth is either controlled by the position of the clay layer or is deeper than the profile depth and therefore cannot be determined.

Comparing the 160-year-old moraine to the youngest moraine we find that the volume density is lower and the surface density is higher in the upper part of the profile. Also unstained areas (colored white in Figure 7) are visible, which indicates that the

285 higher surface density is not caused by the existence of rocks that split up a homogeneously stained area as it was the case for the youngest moraine. In this case there is no total dye coverage of the permeable soil and the preferential flow paths are initialized already near/at the soil surface. The surface density profiles show a decrease in the lower half of the soil profile, which either goes along with a decrease in blue dye coverage and an increase in stone coverage or an increase in blue dye coverage without a significant change in stone coverage. Both indicates a reduction in the amount of separate flow paths and

290 an increase in flow path widths, even if the permeable space is reduced due to an increase in rock content. Also the fraction of stained path widths (SPW) bigger than 200 mm increases, which indicates that the dye plumes widens in deeper soil depths.

Compared to the two youngest moraines the moraine of 3 000 years shows in general a higher surface area density and a lower dye coverage combined with a higher fraction of unstained permeable soil matrix. This indicates that similar to the 160 years old moraine water is transported in individual flow paths, but here there are more flow paths and they have a smaller width.

This can also be seen in the less frequent appearance of stained path width higher than 200 mm (Fig. 7). Similarly to the 160-year-old moraine, preferential flow paths are already initialized at the top of the soil during infiltration (apparent from the white colored areas across the profile in Figure 7).

The oldest moraine with an age of 10 000 years shows the highest surface area density and the lowest volume density combined with the lowest infiltration depths. The surface and volume density profiles show the same pattern: After a peak close to the

300 soil surface, both density profiles show a decrease with soil depth. This means that the dyed water is only transported deeper into the soil via a few individual flow paths.

## 3.3 Flow type classification

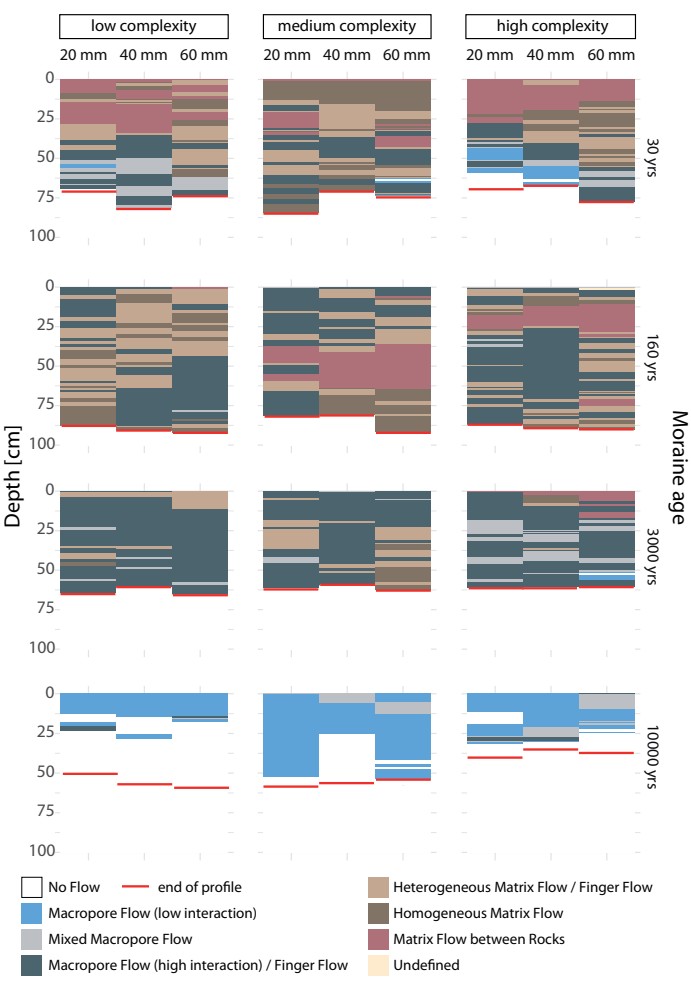

**Figure 9.** Profiles of flow types per age class, vegetation complexity and irrigation amount.

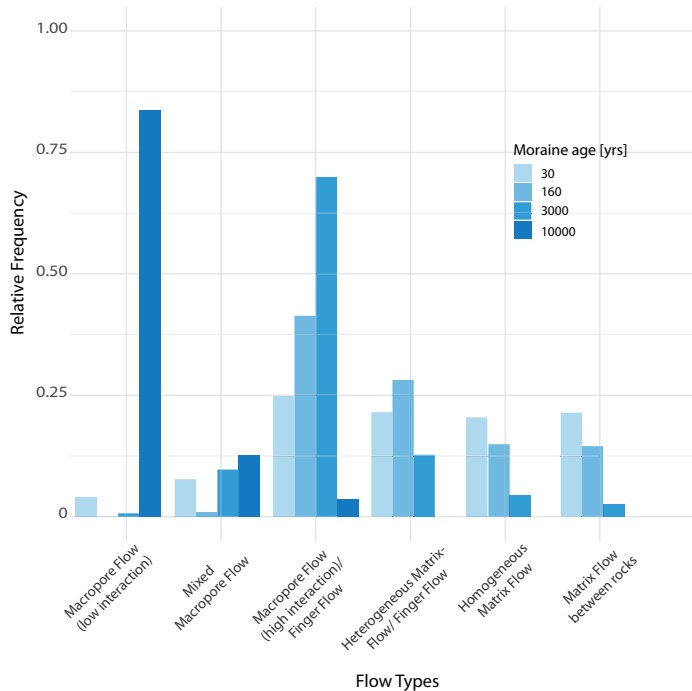

**Figure 10.** Relative frequency distribution of flow types of the four moraine age classes. Basically all observations fit into the six flow type categories. The fraction of observations categorized as "undefined flow type" is negligible.

Using the information in the volume density profiles and the stained path widths to characterize flow types (Weiler, 2001) we found a trend from a rather homogeneous flow pattern with matrix flow in a fast draining coarse textured soil at the youngest
moraine to a more heterogeneous flow pattern with a mix of heterogeneous matrix flow and finger flow at both medium age moraines (Fig. 9). While the flow characteristics of both the 160 and 3 000-year-old moraine are dominated by finger flow with smaller stained path widths, this is much more pronounced in the 3 000-year-old moraine. By contrast, the oldest moraine stands out clearly from the other age groups. Here, only macro pore flow is predominant. For the deeper soil layers, this was confirmed during the field experiments, since the few macro pore flow paths were clearly visible. For the topsoil, the result is
less certain, as the blue areas were very difficult to identify during the image analysis due to the very dark color of the organic layer.

The relative frequency distribution of the flow types per moraine age class derived from the results in Figure 9 show a clear shift in the flow type distribution along the age classes (Fig. 10). At the youngest moraine all types of finger flow and matrix flow are present and the frequency distribution does not show a distinct peak at any flow type. With increasing age macropore
flow becomes more and more important and the peaks in the frequency distribution become more and more pronounced (Fig. 10).

## 4 Discussion

### 4.1 Evolution of soil texture and structure

**The early years: 30-160**

The observation of bulk density, porosity and soil texture show significant differences between the age groups as well as some clear trends with age. The 30-year-old soil is characterized by coarse material, with a soil texture composed of almost three quarters of sand. The soil texture observed at the 160-year-old moraine does not differ strongly from the 30-year old moraine, whereas slight changes are already evident in porosity and bulk density. These findings are similar to findings by Dümig et al. (2011), who found no specific trend in grain size distribution for soils in the range of 15 to 140 years at a soil chronosequence in

the foreland of the retreating Damma glacier (Switzerland). They furthermore found a high variability in grain size distribution within the single age classes. However, in the same study a slight decrease in variability with increasing age and a noticeable higher clay content was found at the reference site with an age of more than 700 years. He and Tang (2008) also revealed a non-linear increase in maximum clay content for soils up to 180 years at a glacier foreland in a monsoon-temperate region in southwest China.

After 160 years of soil development the porosity in the top layer increased and bulk density decreased. In general, these changes could be linked to changes in grain sizes, as the breakdown of particles leads to an increase in total pore space (porosity) and thus to a reduction in bulk density (Arvidsson, 1998). However, since changes in grain sizes were only marginal, the vegetation development, which includes an increase in root activities, litter accumulation, and biological activities in the root zone, is likely the main cause for changes in bulk density and porosity [Neris et al. (2012), Carey et al. (2007)]. The vegetation coverage of

both moraines differs significantly (Maier et al., 2019). The youngest moraine still shows only low vegetation cover with only single plants (mainly grasses and forbs) and little root mass with an observed maximum rooting depth of 15 cm (occasionally up to 30 cm), whereas the 160-year-old moraine already has a relatively closed vegetation cover with a combination of shrubs and smaller plants like forbs and grasses forming a loose root network with roots up to a maximum diameter of 5-6 mm and a maximum depth of 35 cm (as observed during the excavation of the soil profiles).

Similar findings in bulk density evolution were also observed by Crocker and Major (1955) who found a decrease in bulk density over the first 200 years of soil development from more than 1.4 g cm$^{-3}$ to less than 0.8 g cm$^{-3}$ for glacial till in southeastern Alaska. A less pronounced reduction was also found by Crocker and Dickson (1957). He and Tang (2008) found a reduction for the time span of 180 years from appr. 1.42 to 0.95 g cm$^{-3}$ that was also more distinct in the upper horizon. Vilmundardóttir et al. (2014) revealed at a glacier foreland in southeast Iceland under maritime climate conditions a reduction

from 1.36 to 1.07 g cm$^{-3}$ for a time span of 120 years . All studies mentioned above linked this decrease in bulk density to the vegetation development with time.

**Intermediate stage: 3 000 years**

At 3 000 years of soil development we observe a distinct increase in silt and a reduction in sand content and this development continues, as observed in the 10 000 year moraine, where the silt content now makes up the largest share. These findings agree
with findings by Douglass and Bockheim (2006) who studied several moraines in Buenos Aires with ages ranging from 16 000 year to even 1 000 000 years and found an accumulation of clay-sized particles with increasing age, but with a decrease in accumulation rate over the years. A high fraction of silt is very common for soils in mountain areas (Ellis, 1992). Physical weathering due to high fluctuations between day and night temperature and freezing cycles (Birse, 1980) leads to a reduction in grain size, without changing the particle mineralogy (Ellis, 1992).
The soil material at the 30-years-old moraine showed a relatively uniform porosity and bulk density throughout the profile. After 3 000 years, porosity increases and bulk density decreased even further, and this development is now also visible in deeper soil depths. The continuous increase in porosity and reduction in bulk density can be attributed to the continuing change in soil texture on the one hand and on the other hand to the pronounced vegetation development. Especially the latter with the resulting accumulation of soil organic matter (see Fig. 6(c)) and the growth of an even denser root network that is now over 35
360 cm deep, is the main cause for the pronounced changes in the top soil.

**The late stage: 10 000 years**

The oldest moraine shows a significantly higher silt content and porosity compared to the 3 000-year-old moraine and a significantly lower bulk density. The change is visible at all soil depths, with the porosity in the uppermost depth being distinctly higher than the other depths. These differences in soil properties between the soil layers also indicate a progressive formation
of distinct horizons in the soil.
The significantly higher porosity in the upper layer of the oldest moraine is caused by its thick organic layer (thickness up to 20 cm), which is characterized by porosity of up to 90 percent [this was also found by Nyberg (1995) in sandy-silty till on the west coast of Sweden and Carey et al. (2007) in organic soils in a permafrost region in northwest Canada].
Musso et al. (2019) investigated the evolution of pore sizes in the top 5 cm at the same soil chronosequence and found an
370 increase in number of small soil pores and a decrease in relative proportion of macropores (pore diameter > 0.05 mm) between 160 and 10 000 years. Thus the high porosity in the organic top layer at the oldest moraine is mainly composed of small pores. The top layer therefore has an increased water storage and water holding capacity. Due to the finer soil texture and higher porosity the total storage water capacity of the oldest moraine is larger than that of the younger moraines.
An investigation of the saturated hydraulic conductivity evolution of the near-surface (in 0-5, 5-20, and 20-40 cm) at the same
chronosequence by Maier et al. (2019) found a decrease with increasing moraine age and soil depth. Saturated conductivity was found to be negatively correlated with the fraction of fine particles. The decrease in gravel content and the increase in silt seem to have an even a stronger effect on the saturated conductivity than the root network development (Maier et al., 2019).

**Soil heterogeneity and vegetation complexity**

It is well known that soil properties are spatially heterogeneous [Bevington et al. (2016), Hu et al. (2008)]. As it was not possible to account for this variability with a large sample size, i.e. with a large number of experiments, we decided to take a different approach: Assuming that vegetation cover and subsurface flow paths are strongly linked, we took the variability in vegetation cover as a proxy and used it in an attempt to bracket this variability: per moraine three locations that differ in their vegetation complexity (low, medium, high) were chosen for soil sampling and the dye tracer experiments. The analysis of the structural soil properties shows that there is a slight increase in spatial heterogeneity with age, especially in the top soil (increase in interquartile ranges for all properties in the top layer in Fig. 6), but occasionally also individual depths show a higher heterogeneity, irrespective of age.

The flow path analysis differentiated according to the vegetation complexity showed no systematic influence of the complexity level on the results. Heterogeneities within the individual experimental subplots were taken into account by averaging the volume density and surface area density across the five vertical profiles per subplot instead of relying on individual profiles. We therefore assume that the results of the flow path analysis are sufficiently representative to investigate their evolution across the chronosequence.

## 4.2 Evolution of flow paths

The flow type classification by Weiler (2001) was used to classify the volume density patterns into flow type categories. A comparison between the observations made during the excavation and the derived flow types showed that in this case an adaptation of the flow type classification was necessary. On the one hand, this adaptation involved the treatment of rocks to prevent mis-classification and, on the other hand, we introduced the possibility of finger-like flow paths with smaller widths. After these adaptions the derived flow types correspond well with the observations made in the field.

The observed staining patterns and derived flow types show a significant difference between the age groups, whereas no significant difference was observed with respect to vegetation complexity and irrigation amount.

**The early years: 30-160**

At the 30-year-old moraine the water infiltrates homogeneously into the soil, probably due to the very low vegetation coverage and the coarse material texture. The dye pattern showed a mainly homogeneous staining of the soil material, thus derived flow types are mainly matrix flow in form of homogeneous and heterogeneous matrix flow as well as matrix flow between rocks. Also finger flow occurs at the boundary to the clay layer or is caused by large blocks of rock, which are surrounded by clay. The determined macro pore flow takes place only within the clay layer at a depth below 50 cm. In the clay layer, no significant biopores were identifiable, which is why it is assumed that the water is transported in cracks or along material interfaces. The upper coarse soil material with large pores and a low water holding capacity causes the water to be transported quickly deeper into the soil.

After 160 years the derived predominant flow types shift to heterogeneous matrix flow and finger flow. The observed widening

of the dye plumes in deeper soil depths might be caused by a change in material or a reduction of hydrophobicity with soil depth where the influence of plants and organic material decreases (Blume et al., 2009). The dye coverage images show unstained soil areas starting also at the top of the soil profile, which indicates that preferential flow paths are initialized already at the soil surface or in the near-surface layer. This was also observed during the irrigation where we saw that the irrigated water often mainly infiltrated in depressions. Grass patches also tend to inhibit infiltration. It was observed that in the presence of dense grass patches, the water infiltrated only next to the patches, leaving the area below the patches unstained.

## Intermediate stage: 3 000 years

Similar observations were made at the 3 000-year-old moraine. The image analysis revealed that preferential flow paths start at the soil surface and thus are initiated by vegetation and micro topography causing a heterogeneous infiltration pattern. Field observations also revealed that heterogeneous infiltration was not only created due to dense grass patches, but also occured under relatively homogeneous grass cover. A laboratory test of the water drop penetration time [DeBano (1981), Doerr et al. (2000)] on a soil sample of the upper soil material showed that the organic layer is highly water repellent in dry conditions (air dried for 2 weeks, water drop penetration time > 10 minutes). An increase in the Hydrophobicity Index (Tillman et al., 1989) with increasing moraine age was also found by Maier et al. (2019). Thus, we conclude that hydrophobicity of the organic top layer has a big impact on infiltration and the initiation of unstable flow. Unstable flow occurs when horizontal wetting fronts break into fingers or preferential flow paths during the downward movement (Hendrickx and Flury, 2001). Compared to the 160-year-old moraine the 3 000-year-old moraine is characterized by a higher number of narrower preferential flow paths.

The derived dominant flow type class at the 3 000-year-old moraine is macro pore flow with high interaction/finger flow. Of both possible flow types, finger flow is the prevalent flow process causing the dye pattern. Several studies linked the formation of finger-like flow paths to hydrophobic properties of the soil [Wallach and Jortzick (2008), Dekker and Ritsema (2000), Ritsema and Dekker (1994), Blume et al. (2008), Wang et al. (2018), Hardie et al. (2011)]. It is assumed that hydrophobic compounds that are released during the decay of litter (Reeder and Jurgensen, 1979) or by root activity (Doerr et al., 1998) coat soil particles or are deposited in the pore space and thus create a hydrophobic soil matrix (Doerr et al., 2000). The humid and cool climate of former glacial areas leads to a slow decomposition of vegetation and thus to an accumulation of hydrophobic compounds (Doerr et al., 2000).

## The late stage: 10 000 years

At the oldest moraine, we saw a distinctly shallower infiltration depth (Fig. 7). During the experiment no surface runoff was observed. Most of the water was stored in the organic top layer. The soil beneath the top layer was almost completely unstained and water was transported only via a few macropores into deeper layers. A dense network of roots was only observed in the organic top layer, which included the thicker roots of the alpenrose (Rhododendron ferrugineum). The root network in the soil underneath was less dense with roots of smaller diameters but extended to a depth of more than 50 cm. Although the vegetation

cover has been reduced to decrease interception, the interception storage capacity at the oldest moraine is still comparatively high. Thus, a reduction in the water available for infiltration cannot be ruled out.

**Impact of rocks**

The rock content at the 30, 160, and 3 000-year-old moraines is relatively high, with especially large rocks (widths > 20 mm) in deeper parts of the soil (Fig. 5(b) and Fig. 7). The large rocks lead to a reduction of the permeable area and thus can cause funnel flow (Hendrickx and Flury, 2001). This type of preferential flow was especially observed at the youngest moraine, where large boulders (> 25 cm) located at deeper soil depths were surrounded by unstained fine textured material. Smaller sized rocks
in the upper part seemed not to have an influence on water transport (apart from reducing the flow-through volume), since these rocks and the surrounding soil were completely stained.

A splitting of flow paths caused by rocks was also observed a few times at the 160- and 3 000-year old moraines. In this case, water flowing past the sides of medium to large sized rocks create a type of finger flow that is not caused by water repellency or air entrapment. The tendency of higher rock contents to increase the number of flow paths was also found by Bogner et al.
(2014).

**Flow path controls along the age gradient**

Integrating all of our findings on soil structural parameters, texture, vegetation cover and flow path patterns provides an overview over their co-evolution and highlights the derived major flow path controls (Fig. 11).

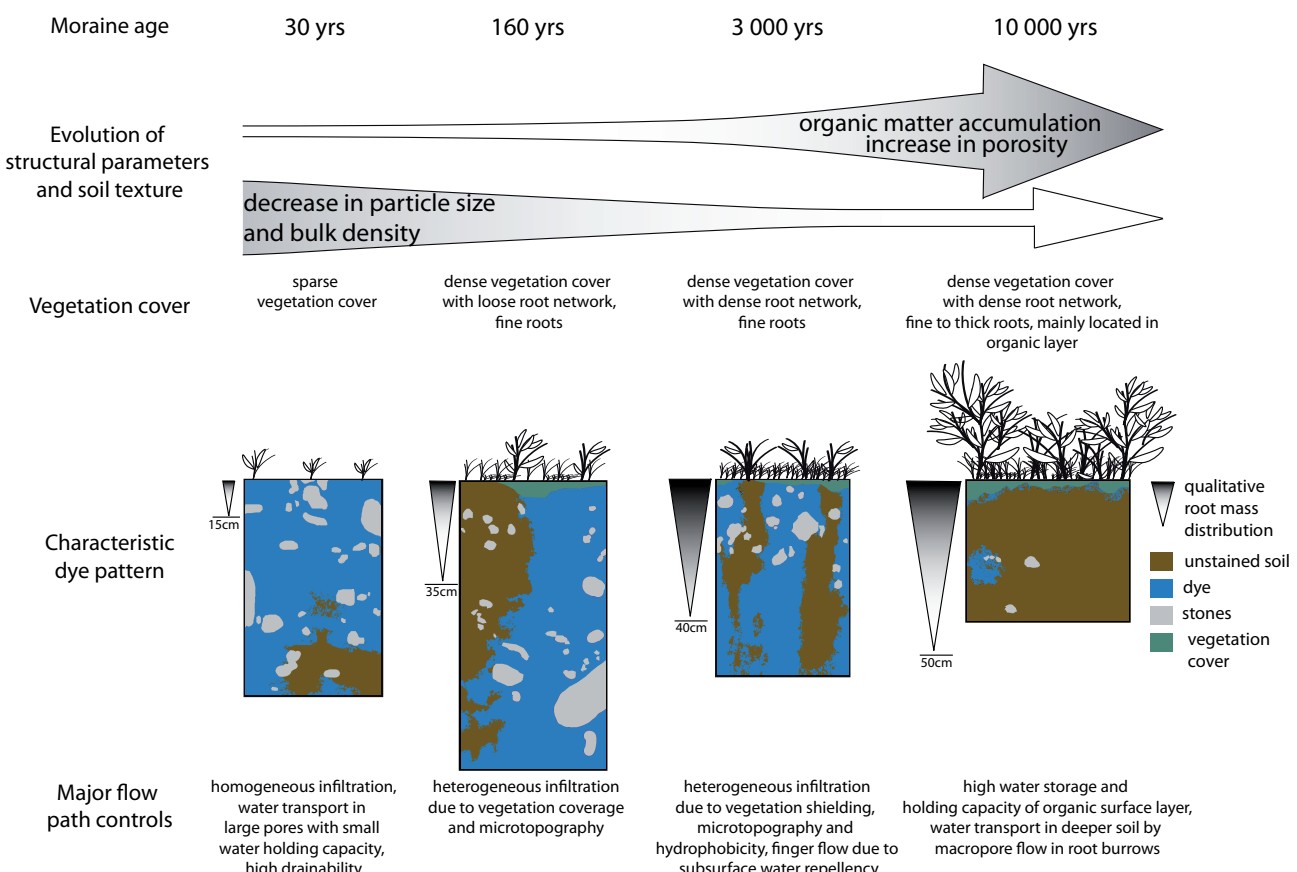

**Figure 11.** Sequence of observed characteristic dye patterns and derived flow path controls compared to the qualitative evolution of soil texture, structural parameters and vegetation. The shade of the root mass-distribution-triangles is a measure for the vertical root mass distribution with darker color indicating a higher root mass. The width of the triangles is a measure for the root mass comparison between the moraine ages, with broader triangles indicating a higher root mass.

Along the co-evolution of soil and vegetation over 10 000 years the major controls of subsurface flow paths change. At the youngest moraine flow paths are only controlled by soil texture. The coarse material leads to downward movement of the infiltration front. Preferential flow paths only occur at the interfaces between coarse and finer material.

At the medium aged moraines flow paths are mainly controlled by vegetation shielding, microtopography and hydrophobicity. The latter is assumed to have an increased impact at the 3 000-year-old moraine. After 10 000 years of hillslope evolution subsurface water transport is highly preferential and controlled by flow paths caused by root channels or boundaries of textural classes. Water storage in the organic layer which is also the main rooting horizon increases strongly.

# 5 Conclusions

Using brilliant blue dye experiments and soil sampling we investigated the evolution of water transport paths along soil forming processes. To our knowledge this is the first study examining flow path evolution across the millennia in such detail. The evolution of the grain size distribution shows that grain size decreases with increasing age. The biggest changes are in the sand and silt fraction. Furthermore, water flow defining structural parameters such as porosity and bulk density change during soil development, resulting in an increasing water storage capacity with age. The depth dependent evolution of these parameters supports our hypothesis that the soil material develops with increasing age from a homogeneously mixed material to a depth differentiated soil system with vertical gradients in flow and storage defining soil properties. Changes in these flow defining parameters are caused by the evolution of grain size distribution and vegetation.

The derived flow types also support our hypothesis that vertical subsurface flow path types and their vertical extent change through the millennia. Flow types change from homogeneous matrix flow in a fast draining coarse textured soil to a heterogeneous matrix and finger flow over the first 100-3 000 years. At very young moraines the water is homogeneously distributed within the soil matrix. However, the water storage capacity is relatively low due to the coarse material and water is transported quickly deeper into the soil due to the high drainability. At the medium age moraines, water is transported preferentially via finger-like flow paths deeper into the soil by leaving parts of the soil dry.

With increasing hillslope age, we expected macropores induced by root activities to become more important. After 10 000 years, were the amount of soil matrix macropores decreased significantly, macro pore flow along roots plays an important role, but is not very pronounced. Only a few roots reach beyond the organic top layer. However, this allows a fast transport of water from the upper layer into deeper soil. The organic top layer has a pronounced influence on the soil water budget, by storing a significant amount of water. The increase in water storage with increasing age of the moraines also caused a reduction in infiltration depth.

The proportion of preferential flow paths increases with soil age. Preferential flow is, however, not only caused by macropores, but especially for the medium-age moraines seems more controlled by soil surface characteristics such as vegetation patches, micro-topography and hydrophobicity. Thus, the evolution of flow paths is tightly linked to the complex interplay of soil forming processes and vegetation development over the millennia. A lot of changes in vegetation cover, soil (hydraulic) properties and flow paths occur within the first 160 years.

It was shown that the complex interaction of vegetation and soil development and its proven effect on flow path development also impacts the water balance, as the storage and conductivity properties of the soil change. However, the interplay between preferential flow paths and the soil hydraulic behavior not only influences the soil water budget, but also runoff formation. These findings provide important insights on hydrological flow path evolution in transient systems.

*Author contributions.* AH and ES conducted the field experiments, prepared the images and performed the analysis. AH collected the soil samples and conducted the laboratory analysis. MW and TB were involved in planning the fieldwork. AH prepared the manuscript with contributions from all co-authors.

*Competing interests.* The authors declare that they have no conflict of interest.

*Acknowledgements.* This work is funded by the German Research Foundation (DFG) and the Swiss National Science Foundation (SNF) within the DFG-SNF-project Hillscape (Hillslope Chronosequence and Process Evolution). We thank Moritz Lesche and Jonas Freymüller for their assistance in the field. We also thank Kraftwerke Oberhasli AG (KWO) for permission to conduct the experiments. Many thanks to Thomas Michel and his team of the Alpin Center Sustenpass and Peter Luchs for their kind hospitality and support.

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
