# Peer review of "Field observations of soil hydrological flow path evolution over 10 Millennia"

_Hydrology and Earth System Sciences, 2020_

## Referee Comment (RC1) · Anonymous Referee #1 · 21 Feb 2020

General Comments In this paper, the authors investigated changes in soil characteristics and water flow through time by examining a chronosequence of soils from a retreating glacier. The study is very thorough, detailed, and makes conclusions that I think are novel and interesting to the community. The paper is mostly very well written and structured, but I have a few areas of concern and/or need for clarification, detailed below.

Specific Comments Issue 1: Hypotheses. I think the hypotheses in lines 9-13 on page 3 could be improved or re-stated as research questions. In general, I think they are a bit vague for hypotheses. For example in (1), what does "change" mean?, in (2) what does "more important" mean? And (3) what process is hypothesized to lead to a reduction in particle size and/or increase in porosity and/or increase in subsurface

water storage? And for (3) should this be more than one question? It hits a few different predictions/questions.

I think the wording used when addressing the hypotheses in the conclusion is also a bit strong. I think there's an argument to be made that it is okay to say "confirmed" about a hypothesis, just being careful to avoid "proved" but it gave me pause. I think the conclusion could benefit from a few statements identifying the uncertainty in the set up and analysis and then caching the "confirmation" of the hypotheses in those terms.

Issue 2: Description of the study design. I had to read through the methods several times, taking notes and adding up samples from "plots" and "subplots" trying to be sure I understood where the data was coming from. I think the section would benefit from a paragraph in 2.2 that makes very clear: how many plots are there in each moraine? How far away are they from each other? (can this also be shown in FIgure 1?) are there subplots in every plot or just the dye application plots? I realize this information is all included in the paper, but it's scattered throughout the methods so some piecing together was required for me to figure it out.

Issue 3: Heterogeneity. I think it'd be good to have more discussion about the heterogeneity in these moraines, then how that heterogeneity was addressed in the study design and how it affects the interpretation of the results. Would you expect these moraines to be pretty homogeneous? If not, how were the heterogeneities accounted for, and how likely is it that the results might be different if the sites were placed differently?

Issue 4: Discussion structure. This discussion does a good job of putting the findings of the paper in context with previous work, but could you also add some information about how these changes are happening? Having processes tied to the changes would be really helpful for applying the findings here to other places. There is a bit of discussion about this with regards to vegetation and flowpaths, but not so much with the texture and structure.

Additionally, in the first part of the discussion findings are sort of point-by-point related to previous literature. I wonder if the readability of the section might be improved by restructuring a bit to talk about how some of the changes in texture and structure happen together rather than breaking them all into separate paragraphs. There are a lot of findings here, and I realize that makes it kind of hard to present them concisely, so it's just a suggestion. But maybe similar processes are leading to the changes observed, and discussing those processes and the results may help.

Technical Corrections Page 6 Line 15 parameter should be plural

Page 6 Line 24 should "amount" be "number"?

Page 7 Line 6 I think the comma after "both" can be removed

Page 7 Lines 22 and 29 can you add a note explaining what you mean by disturbed and undisturbed? I assume the structure was preserved in the undisturbed sample... but they were both removed from the site, so they were definitely disturbed!

Page 16 Line 14 maybe "agree" would be better than "correspond"?

Page 18 Line 17 add "age" after "increasing moraine"

Page 18 Line 17 what is unstable flow?

Page 18 Line 27 maybe use a different word than "significantly" since it isn't a statistical comparison

Page 18 Line 27 see = saw

Page 20 Line 12 remove "also"

Page 20 Line 33 remove "already"

---

## Referee Comment (RC2) · Nicholas Jarvis (Referee) · 5 Mar 2020

This paper presents the results of dye tracing experiments showing how the mechanisms of preferential flow (PF) change with soil development in alpine moraines exposed by glacial retreat. This work culminates in a very nice schematic diagram (Figure 9) that summarizes and illustrates the main findings. The main strength of the paper is that the study of PF from this kind of pedological perspective is still really quite novel. Nevertheless, the authors are not the first researchers to have taken this kind of pedological approach and it would strengthen the paper if some of this relevant earlier work could be mentioned in the Introduction. The authors could check out Quisenberry et al. (1993), Lin (2003), Cammaraat and Kooijman (2009) and Jarvis et al. (2102).

[Figure]

Another interesting and rather novel aspect of the paper is the demonstration of the importance of stones and rocks for generating and maintaining preferential flow. Maybe the authors could also cite Bogner et al. (2014), who demonstrated the same thing. The paper is generally well written and presented and easy to read, although the language could be improved further by a native speaker. One minor concern is that the author's use of terminology related to PF is, at times, unnecessarily confusing. I have two other criticisms. First, the methods are not described in sufficient detail. Secondly, the authors could do a better job of discussing their results with respect to the fundamental processes causing the observed changes in flow patterns. These aspects are explained more fully in the following.

Methods: It was not clear to me what aspects of the vegetation cover were actually measured. For example, you use the term "mapped" on page 4 at line 15, but this is a rather vague. Do you have measurements of anything like above-ground biomass or was only species composition recorded? Please write this more explicitly. The description of vegetation complexity on page 5 at lines 2-4 is also not very helpful. Can you give a brief description here, maybe with an equation? The reader should not have to consult another paper (Musso et al.).

The description of the irrigation procedure on page 5 (lines 15-18) was quite difficult to follow. It seems as if the irrigation pattern was different between the plots. Why was that? Perhaps a schematic figure might help to explain this.

The method to classify the flow patterns into different groupings is described only very briefly (on pages 6/7) and it was also difficult to follow. To complement the text (e.g. at lines 28-30 on page 6), could you give an equation or perhaps include a schematic diagram (or both)? This procedure is quite central to the paper, so it is important to explain this carefully.

The description of the particle size analysis on page 7 makes no mention of the gravel/stone fraction (>2 mm). Did you measure the content of stones/gravel? I know

this is very difficult in stony soils, so it is understandable if you didn't, but I think this should be stated.

The description of how bulk density and porosity were measured (page 7, lines 31-32) is quite vague. Can you describe more exactly (but still briefly) how you measured bulk density and (especially) porosity? It would be good to give some details, because the porosity values are extremely small in the young moraine and in the subsoils. I suppose this is because of the high stone content, but it could also be because air got trapped in the samples during saturation.

The authors do not report any measurements of soil organic matter content (SOM). This data should ideally be included in the paper, as the build-up of SOM over millennia due to the growth of vegetation seems to be a very important control on the observed changes in the flow patterns. If SOM was not measured, then I think it should be measured now and the results included in the paper (the analysis is quick and cheap).

Processes: Can you explain (e.g. on page 16, lines 9-17) why the texture becomes finer with age? Is it due to weathering or is it deposition of fine materials by wind, or maybe both (or something else)? The cause(s) might be obvious to the authors, but perhaps they will not be to all readers.

The process(es) operating to increase porosity and decrease bulk density should be explained better (e.g. on page 16, line 20, and lines 33-34). I presume that it is mostly related to a build-up of organic matter in the soil, which is supplied by the litter and roots of the increasingly dense vegetation cover and subsequently processed by soil micro-organisms and fauna, which ultimately results in a more open (aggregated) soil structure.

The authors associate the homogeneous flow patterns found in the young moraine with "gravity-driven" water flow (e.g. on page 15, line 2; page 18, line 2; page 20, line 21). This is rather misleading to my mind. Fundamentally, it must be the case that both gravity and capillarity were driving the infiltration process in all your experiments,

because the soils were (presumably) initially quite dry. In fact, the authors do not really need to discuss whether gravity or capillarity dominated the flow patterns in the different moraines, but if they want to do so, then I think in reality, it is the opposite of what they write. Both macropore flow and finger flow are gravity-dominated processes, whereas a homogeneous flow pattern implies that capillarity was strong enough to prevent the development of any lateral non-equilibrium in soil water pressures. It is this lateral non-equilibrium in water pressures during flow that is a fundamental characteristic of PF.

Confusion over terminology: Considering the underlying physical mechanisms, there are three main types of PF (macropore flow, finger flow and heterogeneous flow) and this is indeed the basis of the classification system that the authors make use of in the paper. However, the authors unnecessarily introduce some confusion at a couple of places in the paper by referring to another classification scheme, one that is not especially useful in my opinion:

i.) page 2 (lines 18/20): There is no good reason to distinguish crack flow from burrow flow (does burrow flow include flow in channels created by root decay?). These can all be lumped into macropore flow (as you do later). If you want to define some sub-groups according to the origin of macropores, you should talk about flow in biopores (which includes both root and faunal channels) not burrow flow.

ii.) page 17, line 34: "In the clay layer, no significant macropores were identifiable, which is why it is assumed that the water is transported in cracks . . ..". Cracks are also macropores. You should replace the term macropores by biopores.

Corrections

1. The text at the end of the Introduction should be re-arranged. The hypotheses at lines 6-11 don't make much sense at the moment, because they are specific to the case of glacial moraines. It's not clear to the reader where these hypotheses come from. If you move this text to line 19 (after ". . .impacts water flow paths"), I think it will

make more sense, especially if you add "... in glacial moraines in the Swiss Alps" after "...landscape evolution", and delete the last sentence in the first paragraph.

2. Abstract, Line 1: you should delete "The presence or absence of ..."

3. page 3, line 1: add "volcanic" after "...younger"

4. page 4, lines 15-16: delete "by the project partners .... Germany"

5. Page 7, line 10: maybe you could add "..... and flow mechanisms" after "different properties"

6. Page 8, line 1: add ".... moraines of differing ..." after "four"

7. Page 8, line 6: replace "the entire" by "all"

8. Page 8, figure 3 caption: I presume that these results are % of the fine earth fraction (< 2mm). It would be good to state this here.

9. Page 10, line 5: I don't think you should talk about hillslopes as you haven't mentioned anything about site topography. You could just replace "hillslope" here by "moraine"

10. Page 12, line 7: This is ambiguous, but I think you mean: "For all four moraines, the volume density is largest in the top half of the soil profile"

11. Page 13, line 2: interpreting dye tracing patterns can be tricky, since you only get a snapshot in time of a dynamic process. In this particular case, I think it's possible that even if the staining was homogeneous, it doesn't necessarily mean that PF didn't occur. PF could have occurred from the soil surface, but the signs of this may have been obliterated by the later (slower) downward movement of a uniform wetting front in the soil matrix. I am not saying that this is what happened (I'm confident that your interpretation is correct), but I think you could recognize this possibility.

12. Page 16, lines 26-27: I don't understand how the decrease in bulk density in the

first 160 years can be related to a change of particle sizes, since this was marginal. It must be primarily due to the increase in SOM content.

13. Page 16, lines 26-34: there is no need to have separate discussions for porosity and bulk density, because they are very closely linked (via the particle density). You could simplify and shorten the text between lines 18 and 34: you only need to write that the increase of porosity and decrease of bulk density was presumably a result of organic matter build-up in the soil due to the development of a denser vegetation cover.

14. Page 17, lines 2-3: it should be briefly explained (with a supporting reference) how the change in texture could affect bulk density. Presumably the finer particles fill the spaces between the coarser particles? However, I think that the effects of texture on bulk density are usually considered to be relatively small. I think that the increase in SOM content (and associated biological activity in the soil) must be the main reason for the decrease in bulk density.

15. Page 17, line 20: "texture" in this context is quite a vague term. Was it clay content? Please be more explicit.

16. Page 17, line 30: the coarse nature of the material must be important too?

17. Page 18, line 27: replace "lower" by "shallower"

18. Page 18, line 32: should be: ". . .. cover was removed to decrease . . .."

References

Bogner, C. et al. 2014. Quantifying the morphology of flow patterns in landslide-affected and unaffected soils. Journal of Hydrology, 511, 460–473.

Cammeraat, E.., Kooijman, A., 2009. Biological control of pedological and hydrogeo-morphological processes in a deciduous forest ecosystem. Biologia 64, 428–432.

Jarvis, N.J., Moeys, J., Koestel, J., Hollis, J.M. 2012. Preferential flow in a pedological perspective. In: Hydropedology: synergistic integration of soil science and hydrology

(ed. H. Lin), Academic Press, Elsevier B.V., pp.75-120.

Lin, H. 2003. Hydropedology: bridging disciplines, scales, and data. Vadose Zone J., 2, 1-11.

Quisenberry, V., Smith, B., Phillips, R.E., Scott, H., Nortcliff, S. 1993. A soil classification system for describing water and chemical transport. Soil Sci. 156, 306–315.

---

## Author Comment (AC1) · 22 Mar 2020

**Response to Reviewer comments**

**Response to Reviewer 1**

**General Comments**

In this paper, the authors investigated changes in soil characteristics and water flow through time by examining a chronosequence of soils from a retreating glacier. The study is very thorough, detailed, and makes conclusions that I think are novel and interesting to the community. The paper is mostly very well written and structured, but I have a few areas of concern and/or need for clarification, detailed below.

**Response to General Comments**

The authors would like to thank the reviewer for spending his/her time to review and make valuable comments to improve our manuscript. We will address these comments and suggestions below.

**Specific Comments Issue 1:**

Hypotheses. I think the hypotheses in lines 9-13 on page 3 could be improved or re-stated as research questions. In general, I think they are a bit vague for hypotheses. For example in (1), what does "change" mean?, in (2) what does "more important" mean? And (3) what process is hypothesized to lead to a reduction in particle size and/or increase in porosity and/or increase in subsurface water storage? And for (3) should this be more than one question? It hits a few different predictions/questions.
I think the wording used when addressing the hypotheses in the conclusion is also a bit strong. I think there's an argument to be made that it is okay to say "confirmed" about a hypothesis, just being careful to avoid "proved" but it gave me pause. I think the conclusion could benefit from a few statements identifying the uncertainty in the set up and analysis and then caching the "confirmation" of the hypotheses in those terms.

**Response to Specific Comments Issue 1:**

We agree and will specify and rephrase our hypotheses to:

*More specifically, we test the hypotheses that (1) Vertical subsurface flow path types and vertical extent change through the millennia as: (2) The proportion of macropore flow will increase due to the development of biopores, (3) The soil develops from a homogeneously mixed material into a depth differentiated soil system, and*
*(4) Physical weathering leads to a reduction in particle size and an increase in porosity.*

**Specific Comments Issue 2:**

Description of the study design. I had to read through the methods several times, taking notes and adding up samples from "plots" and "subplots" trying to be sure I understood where the data was coming from. I think the section would benefit from a paragraph in 2.2 that makes very clear: how many plots are there in each moraine? How far away are they from each other? (can this also be shown in Figure 1?) are there subplots in every plot or just the dye application plots? I realize this information is all included in the paper, but it's scattered throughout the methods so some piecing together was required for me to figure it out.

**Response to Specific Comments Issue 2:**

We agree with this comment and will restructure this paragraph on the plot selection, soil sampling and subdivision of the plots during the irrigation experiment to make it more clear that at each moraine three experimental plots were selected, which differ in vegetation complexity (low, medium, high) and that sets of soil samples were taken close to each of these three plots, but are presented as a single set

for the entire moraine. For the irrigation experiments each plot was further divided into three subplots for the application of three individual irrigation amounts. Also the image analysis was done individually for each of the subplots. We will also make it clear in the caption of Figure 1 that we here show only one of the 3 plots per moraine.

It is difficult to provide an overview of the plot locations at all moraines, since the plots were located several meters (10-100m) apart. But we will include this information in the text. To clarify the study design further, we will include a sketch that visualizes the experimental design of the field campaign and contains the information about the plot selection and subdivision as well as the soil sampling scheme (see below).

[Figure]

Specific Comments Issue 3:
Heterogeneity. I think it'd be good to have more discussion about the heterogeneity in these moraines, then how that heterogeneity was addressed in the study design and how it affects the interpretation of the results. Would you expect these moraines to be pretty homogeneous? If not, how were the heterogeneities accounted for, and how likely is it that the results might be different if the sites were placed differently?

Response to Specific Comments Issue 3:
We agree with this comment and will add at the end of the Discussion of the Evolution of soil texture and structure the following paragraph to address this issue:
*It is well known that soil properties are spatially heterogeneous [Bevington et al. (2016) Hu et al. (2008)]. As it was not possible to account for this variability with a large sample size, i.e. with a large number of experiments, we decided to take a different approach: Assuming that vegetation cover and subsurface*

*flow paths are strongly linked, we took the variability in vegetation cover as a proxy and used it in an attempt to bracket this variability: per moraine three locations that differ in their vegetation complexity (low, medium, high) were chosen for soil sampling and the dye tracer experiments. The analysis of the structural soil properties shows that there is a slight increase in spatial heterogeneity with age, especially in the top soil (increase in interquartile range in Fig. 4), but also individual depths at different age classes show occasionally a higher heterogeneity.*

*The flow path analysis differentiated according to the vegetation complexity showed no systematic influence of the complexity level on the results. Heterogeneities within the individual experimental subplots were considered by averaging the volume density and surface area density profiles of the five vertical profiles per subplot. We therefore assume that the results of the flow path analysis are representative and account for the heterogeneity of the moraines.*

Specific Comments Issue 4:

Discussion structure. This discussion does a good job of putting the findings of the paper in context with previous work, but could you also add some information about how these changes are happening? Having processes tied to the changes would be really helpful for applying the findings here to other places. There is a bit of discussion about this with regards to vegetation and flowpaths, but not so much with the texture and structure. Additionally, in the first part of the discussion findings are sort of point-by-point related to previous literature. I wonder if the readability of the section might be improved by restructuring a bit to talk about how some of the changes in texture and structure happen together rather than breaking them all into separate paragraphs. There are a lot of findings here, and I realize that makes it kind of hard to present them concisely, so it's just a suggestion. But maybe similar processes are leading to the changes observed, and discussing those processes and the results may help.

Response to Specific Comments Issue 4:

We agree with this comment and will add a paragraph on the processes affecting the soil texture and structure:

*A high fraction of silt is very common for soils in mountain areas (Ellis, 1992). Physical weathering due to high fluctuations between day and night temperature and freezing cycles (Birse, 1980) leads to a reduction in grain size, without changing the particle mineralogy (Ellis, 1992).*

*The break down in grain size also influences the bulk density and porosity of the soil. Whereas sandy soils have a high bulk density, the breakdown of particles leads to an increase in total pore space (porosity) and thus to a reduction in bulk density. Soils enriched in organic matter content also have a lower bulk density [Neris et al. (2012), Carey et al. (2007)].*

We will also restructure the discussion to improve readability.

Technical Corrections

Page 6 Line 15 parameter should be plural

We agree

Page 6 Line 24 should "amount" be "number"?

We agree

Page 7 Line 6 I think the comma after "both" can be removed

We agree

Page 7 Lines 22 and 29 can you add a note explaining what you mean by disturbed and undisturbed? I assume the structure was preserved in the undisturbed sample…but they were both removed from the site, so they were definitely disturbed!

We will include clarifications in section *2.4 Soil sampling and laboratory analysis* to point out that disturbed samples are samples taken without maintaining the natural soil structure and undisturbed samples were taken with sample rings to provide undisturbed cores which maintain the natural soil structure.

Page 16 Line 14 maybe "agree" would be better than "correspond"?

We agree

Page 18 Line 17 add "age" after "increasing moraine"

We agree

Page 18 Line 17 what is unstable flow?

Unstable flow occurs when wetting fronts start out as horizontal wetting fronts that, under certain conditions, break into fingers or preferential flow paths as the front moves downwards (Hendrickx, 2001) For clarification we will include the sentence and the reference.

Page 18 Line 27 maybe use a different word than "significantly" since it isn't a statistical Comparison We agree

Page 18 Line 27 see = saw

We agree

Page 20 Line 12 remove "also"

We agree

Page 20 Line 33 remove "already"

We agree

---

## Author Comment (AC2) · 22 Mar 2020

**Response to Reviewer comments**

**Response to Reviewer 2, Nicholas Jarvis**

**General Comments**

This paper presents the results of dye tracing experiments showing how the mechanisms of preferential flow (PF) change with soil development in alpine moraines exposed by glacial retreat. This work culminates in a very nice schematic diagram (Figure 9) that summarizes and illustrates the main findings. The main strength of the paper is that the study of PF from this kind of pedological perspective is still really quite novel.

Nevertheless, the authors are not the first researchers to have taken this kind of pedological approach and it would strengthen the paper if some of this relevant earlier work could be mentioned in the Introduction. The authors could check out Quisenberry et al. (1993), Lin (2003), Cammaraat and Kooijman (2009) and Jarvis et al. (2102).

Another interesting and rather novel aspect of the paper is the demonstration of the importance of stones and rocks for generating and maintaining preferential flow. Maybe the authors could also cite Bogner et al. (2014), who demonstrated the same thing.

The paper is generally well written and presented and easy to read, although the language could be improved further by a native speaker. One minor concern is that the author's use of terminology related to PF is, at times, unnecessarily confusing. I have two other criticisms. First, the methods are not described in sufficient detail. Secondly, the authors could do a better job of discussing their results with respect to the fundamental processes causing the observed changes in flow patterns. These aspects are explained more fully in the following.

**Response to General Comments**

We thank the reviewer for spending his time to review and improve our manuscript. We will address all concerns and suggestions below.

We appreciate the references to further literature and will include them as suggested by the reviewer.

**Specific Comments Methods:**

Methods: It was not clear to me what aspects of the vegetation cover were actually measured. For example, you use the term "mapped" on page 4 at line 15, but this is a rather vague. Do you have measurements of anything like above-ground biomass or was only species composition recorded? Please write this more explicitly. The description of vegetation complexity on page 5 at lines 2-4 is also not very helpful. Can you give a brief description here, maybe with an equation? The reader should not have to consult another paper (Musso et al.).

**Response to Specific Comments:**

We agree with this suggestion and will add explanations on how the species composition and vegetation cover was recorded.

We also agree with the comment on the description on vegetation complexity and will add this explanation:

*Three study plots were selected at each moraine, based on degree of vegetation complexity (low, medium and high complexity). Vegetation complexity is characterized by vegetation coverage, number of species and the plant functional diversity. The functional diversity is calculated based on specific leaf area, nitrogen content, leaf dry matter content, Raunkiaérs life form, seed mass, clonal growth organ, root type and growth form. The collection of the required data and calculation of the vegetation complexity was done by the Geobotany Group of the University of Freiburg and is described in more detail in Maier et al. (2019).*

Please note, that we will move this part from chapter 2.1 to chapter 2.2 Field experiments.

The description of the irrigation procedure on page 5 (lines 15-18) was quite difficult to Follow. It seems as if the irrigation pattern was different between the plots. Why was that? Perhaps a schematic figure might help to explain this.

Response to Specific Comments:
We agree with this comment and will clarify the procedure. We will restructure the entire paragraph on plot selection and subdivision of the plots during the irrigation experiment to make it more clear that at each moraine three experimental plots were selected, which differ in vegetation complexity (low, medium, high). For the irrigation experiments each plot was further divided into three subplots for the application of three individual irrigation amounts. We will describe the irrigation procedure as follows:

*The tracer was applied with a hand-operated sprayer connected to a battery powered pump which guaranteed a constant pressure for a uniform flow rate of 60 l/h. For a time-efficient irrigation of the three subplots with three irrigation amounts (20, 40, and 60 mm) and an intensity of 20 mm/h, the irrigation procedure was divided into three steps. In the first step all three subplots were irrigated simultaneously for 60 minutes in a sequence of 5 minutes irrigation and 5 minutes break.*
*This irrigation procedure provides an application of 20 mm to all three subplots. After finishing the first step the first subplot was covered to avoid any additional water input. In the second step, the other two subplots were simultaneously irrigated for additional 60 min in a sequence of 5 min irrigation and 10 min breaks. This irrigation procedure provides an application of additional 20 mm to the two remaining subplots.*
*In the last step only the third subplot was irrigated for 60 min in a sequence of 2 min irrigation and 10 min breaks while the other two plots remained covered providing an additional 20 mm to this last remaining subplot. After the end of tracer application, the entire plot was covered to avoid any disturbance by natural rainfall.*

We will also split up Figure 2 in two Figures and include the illustration of the dye tracer plot and the stepwise irrigation scheme into a separate Figure that shows the experimental set up of the field campaign in more detail:

[Figure]

**4 moraines:** Moraine age: 30 years | Moraine age: 160 years | Moraine age: 3 000 years | Moraine age: 10 000 years

**3 plots per moraine:**

Experimental design at each moraine:

low vegetation complexity ● ▲ | medium vegetation complexity ● ▲ | high vegetation complexity ● ▲

Dye tracer experiment at each plot:

blue dye application

|  |  | 20 mm | 40 mm | 60 mm |
|---|---|---|---|---|
| Irrigation procedure: | hour 1 | 20 mm | 20 mm | 20 mm |
|  | hour 2 |  | 20 mm | 20 mm |
|  | hour 3 |  |  | 20 mm |

**3 subplots per plot:**

Subplot | Subplot | Subplot

50 cm
150 cm
100 cm

● Soil sampling location 1:

|  | Undisturbed samples: | | Disturbed samples: |
|---|---|---|---|
| Depth: | 100 cm³ | 250 cm³ |  |
| 10 cm | 1 | 2 | 1 |
| 30 cm | 1 | 2 | 1 |
| 50 cm | 2 | 1 | 1 |

▲ Soil sampling location 2:

|  | Undisturbed samples: | | Disturbed samples: |
|---|---|---|---|
| Depth: | 100 cm³ | 250 cm³ |  |
| 10 cm | - | 2 | 1 |
| 30 cm | - | 2 | 1 |
| 50 cm | 1 | - | 1 |

The method to classify the flow patterns into different groupings is described only very briefly (on pages 6/7) and it was also difficult to follow. To complement the text (e.g. at lines 28-30 on page 6), could you give an equation or perhaps include a schematic diagram (or both)? This procedure is quite central to the paper, so it is important to explain this carefully.

Response to Specific Comments:
We agree with your suggestion and will include a ternary diagram according to Weiler (2001) (see below) that shows which flow type is assigned to which proportions of the 3 SPW classes in terms of volume density.

[Figure]

The description of the particle size analysis on page 7 makes no mention of the gravel/stone fraction (>2 mm). Did you measure the content of stones/gravel? I know this is very difficult in stony soils, so it is understandable if you didn't, but I think this should be stated.

Response to Specific Comments:
We also measured gravel/stone fraction >2mm, but did not show these results. We will report these numbers in the revised manuscript. Stone coverage in our profiles is furthermore shown in Figure 5

[Figure]

The description of how bulk density and porosity were measured (page 7, lines 31-32) is quite vague. Can you describe more exactly (but still briefly) how you measured bulk density and (especially) porosity? It would be good to give some details, because the porosity values are extremely small in the young moraine and in the subsoils. I suppose this is because of the high stone content, but it could also be because air got trapped in the samples during saturation.

Response to Specific Comments:
We agree with this comment and will include some further explanation on the methods used.

The porosity was determined in the lab using the water saturation method and weighing the samples at saturation and after drying at 105 °C. For saturation, the samples were placed in a small basin. The water level in the basin was increased stepwise by 1 cm per day. When the water level reached the top of the

soil sample and the sample was fully saturated, the bottom of the sample was sealed and the weight at saturation was measured. Bulk density was determined by relating the dry mass after drying at 105 °C to the sample volume.

The authors do not report any measurements of soil organic matter content (SOM). This data should ideally be included in the paper, as the build-up of SOM over millennia due to the growth of vegetation seems to be a very important control on the observed changes in the flow patterns. If SOM was not measured, then I think it should be measured now and the results included in the paper (the analysis is quick and cheap).

Response to Specific Comments:
We will include the SOM data in the revised manuscript:

[Figure]

Specific Comments Processes:
Processes: Can you explain (e.g. on page 16, lines 9-17) why the texture becomes finer with age? Is it due to weathering or is it deposition of fine materials by wind, or maybe both (or something else)? The cause(s) might be obvious to the authors, but perhaps they will not be to all readers.

Response to Specific Comments:
We agree and will include some further clarification that we believe that mainly *Physical weathering due to high fluctuations between day and night temperature and freezing cycles (Birse, 1980) leads to a reduction in grain size down to silt, without changing the particle mineralogy (Ellis, 1992).*

The process(es) operating to increase porosity and decrease bulk density should be explained better (e.g. on page 16, line 20, and lines 33-34). I presume that it is mostly related to a build-up of organic matter in the soil, which is supplied by the litter and roots of the increasingly dense vegetation cover and subsequently processed by soil micro-organisms and fauna, which ultimately results in a more open (aggregated) soil structure.

Response to Specific Comments:
We agree and will include a sentence on that the linkage between the decrease in bulk density and vegetation development is caused by root activities, litter accumulation, and biological activities in the root zone.

The authors associate the homogeneous flow patterns found in the young moraine with "gravity-driven" water flow (e.g. on page 15, line 2; page 18, line 2; page 20, line 21). This is rather misleading to my mind. Fundamentally, it must be the case that both gravity and capillarity were driving the infiltration process in all your experiments, because the soils were (presumably) initially quite dry. In fact, the authors do not really need to discuss whether gravity or capillarity dominated the flow patterns in the different moraines, but if they want to do so, then I think in reality, it is the opposite of what they write. Both macropore flow and finger flow are gravity-dominated processes, whereas a homogeneous flow pattern implies that capillarity was strong enough to prevent the development of any lateral non-equilibrium in soil water pressures. It is this lateral non-equilibrium in water pressures during flow that is a fundamental characteristic of PF.

Response to Specific Comments:
We will remove the reference to gravity-driven flow. However, soils were generally quite wet during the entire field campaign as the total rainfall amount in this region is quite high and hence the soils never really dry out.

Specific Comments Confusion over terminology:
Confusion over terminology: Considering the underlying physical mechanisms, there are three main types of PF (macropore flow, finger flow and heterogeneous flow) and this is indeed the basis of the classification system that the authors make use of in the paper.
However, the authors unnecessarily introduce some confusion at a couple of places in the paper by referring to another classification scheme, one that is not especially useful in my opinion:
i.) page 2 (lines 18/20): There is no good reason to distinguish crack flow from burrow flow (does burrow flow include flow in channels created by root decay?). These can all be lumped into macropore flow (as you do later). If you want to define some subgroups according to the origin of macropores, you should talk about flow in biopores (which includes both root and faunal channels) not burrow flow.

ii.) page 17, line 34: "In the clay layer, no significant macropores were identifiable, which is why it is assumed that the water is transported in cracks …..". Cracks are also macropores. You should replace the term macropores by biopores.

We agree with this comment and will correct our terminology accordingly.

Specific Comments Corrections:
1. The text at the end of the Introduction should be re-arranged. The hypotheses at lines 6-11 don't make much sense at the moment, because they are specific to the case of glacial moraines. It's not clear to the reader where these hypotheses come from. If you move this text to line 19 (after " … impacts water flow paths"), I think it will make more sense, especially if you add "… in glacial moraines in the Swiss Alps" after "… landscape evolution", and delete the last sentence in the first paragraph.
We agree
2. Abstract, Line 1: you should delete "The presence or absence of …"
We agree
3. page 3, line 1: add "volcanic" after "…younger"
We agree
4. page 4, lines 15-16: delete "by the project partners … Germany"
We agree
5. Page 7, line 10: maybe you could add "… and flow mechanisms" after "different properties"

We agree

6. Page 8, line 1: add "…. moraines of differing …" after "four"

We agree

7. Page 8, line 6: replace "the entire" by "all"

We agree

8. Page 8, figure 3 caption: I presume that these results are % of the fine earth fraction (< 2mm). It would be good to state this here.

We agree

9. Page 10, line 5: I don't think you should talk about hillslopes as you haven't mentioned anything about site topography. You could just replace "hillslope" here by "moraine"

We agree

10. Page 12, line 7: This is ambiguous, but I think you mean: "For all four moraines, the volume density is largest in the top half of the soil profile"

We agree

11. Page 13, line 2: interpreting dye tracing patterns can be tricky, since you only get a snapshot in time of a dynamic process. In this particular case, I think it's possible that even if the staining was homogeneous, it doesn't necessarily mean that PF didn't occur. PF could have occurred from the soil surface, but the signs of this may have been obliterated by the later (slower) downward movement of a uniform wetting front in the soil matrix. I am not saying that this is what happened (I'm confident that your interpretation is correct), but I think you could recognize this possibility.

We agree and will add a sentence to account for this.

12. Page 16, lines 26-27: I don't understand how the decrease in bulk density in the first 160 years can be related to a change of particle sizes, since this was marginal. It must be primarily due to the increase in SOM content.

We agree that this sentence is misleading. We already stated in Line 19 that the changes in porosity cannot be traced back to the changes in particle sizes and the same accounts for bulk density.
The sentence is a general statement that reductions in bulk density can be caused by changes in particle sizes and organic matter accumulation. We agree that we need to clarify that at this particular age class accumulation of organic matter is the primary cause for the reduction in bulk density.

13. Page 16, lines 26-34: there is no need to have separate discussions for porosity and bulk density, because they are very closely linked (via the particle density). You could simplify and shorten the text between lines 18 and 34: you only need to write that the increase of porosity and decrease of bulk density was presumably a result of organic matter build-up in the soil due to the development of a denser vegetation cover.

We agree

14. Page 17, lines 2-3: it should be briefly explained (with a supporting reference) how the change in texture could affect bulk density. Presumably the finer particles fill the spaces between the coarser particles? However, I think that the effects of texture on bulk density are usually considered to be relatively small. I think that the increase in SOM content (and associated biological activity in the soil) must be the main reason for the decrease in bulk density.

We agree with this comment and will include some further details that the break down in grain size also influences the bulk density and porosity of the soil. The breakdown of particles leads to an increase in total pore space (porosity) and thus to a reduction in bulk density. But we will also state, that the strong decrease in the topsoil is mainly caused by the increase in SOM.

15. Page 17, line 20: "texture" in this context is quite a vague term. Was it clay content? Please be more explicit.
We agree. We will change the sentence to: *"Saturated conductivity was found to be negatively correlated with the fraction of fine particles. The decrease in gravel content and the increase in silt seem to have an even a stronger effect on the saturated conductivity than the root network development (Maier et al., 2019)."*

16. Page 17, line 30: the coarse nature of the material must be important too?
We agree

17. Page 18, line 27: replace "lower" by "shallower"
We agree

18. Page 18, line 32: should be: "… cover was removed to decrease …"
We agree

---

## Author Response (AR1)

Dear Editor,

We would like to thank you for giving us the opportunity to revise our manuscript. We have revised the manuscript entitled "Field observations of soil hydrological flow path evolution over 10 Millennia" in response to your and the reviewers' comments. Please find attached a revised version of this manuscript as well as a detailed list of our responses to these comments.

We are grateful to you and the reviewers for your interest in our paper and for the detailed evaluation, valuable suggestions, and recommendations. As you will see when examining our revision, the reviewers' comments and recommendations were considered seriously and thoroughly addressed in our revised paper. The four main changes that we have made are as follows:

1. We rephrased our hypothesis to be clear and more specific.
2. We improved the description of the methods.
3. We improved the structure of the discussion.
4. We corrected our terminology to preferential flow paths

Additionally, we have addressed all other comments and suggestions.

We further extended section 2.3. Image analysis by a processing step that was unintentionally not included in the first manuscript version. We therefore updated former Figure 2 (now Figure 3) and included the missing explanations in line 167 to 170:

*Due to poor lighting conditions or a heterogeneous background color distribution in the soil caused by material transitions, small stones or organic matter, the image analysis software was not able to recognize all large dye stains as coherent objects. Thus, a manual correction of the images using the photographs was necessary (see Fig 3).*

**Response to Reviewer comments**

**Response to Reviewer 1**

**General Comments**
In this paper, the authors investigated changes in soil characteristics and water flow through time by examining a chronosequence of soils from a retreating glacier. The study is very thorough, detailed, and makes conclusions that I think are novel and interesting to the community. The paper is mostly very well written and structured, but I have a few areas of concern and/or need for clarification, detailed below.

**Response to General Comments**
The authors would like to thank the reviewer for spending his/her time to review and make valuable comments to improve our manuscript. We will address these comments and suggestions below.

**Specific Comments Issue 1:**
Hypotheses. I think the hypotheses in lines 9-13 on page 3 could be improved or re-stated as research questions. In general, I think they are a bit vague for hypotheses. For example in (1), what does "change" mean?, in (2) what does "more important" mean? And (3) what process is hypothesized to lead to a reduction in particle size and/or increase in porosity and/or increase in subsurface water storage? And for (3) should this be more than one question? It hits a few different predictions/questions.
I think the wording used when addressing the hypotheses in the conclusion is also a bit strong. I think there's an argument to be made that it is okay to say "confirmed" about a hypothesis, just being careful to avoid "proved" but it gave me pause. I think the conclusion could benefit from a few statements identifying the uncertainty in the set up and analysis and then caching the "confirmation" of the hypotheses in those terms.

**Response to Specific Comments Issue 1:**
We agree and specified and rephrased our hypotheses in lines 75 to 80 to:

*Therefore, this study addresses the occurrence and the evolution of preferential flow during the first 10000 years of landscape evolution in glacial moraines in the Swiss Alps. More specifically, we test the hypotheses that (1) Vertical subsurface flow path types and vertical extent of flow paths change through the millennia as: (2) The proportion of macropore flow will increase due to the development of biopores, (3) The soil develops from a homogeneously mixed material into a depth differentiated soil system, and (4) Physical weathering leads to a reduction in particle size and an increase in porosity.*

Specific Comments Issue 2:
Description of the study design. I had to read through the methods several times, taking notes and adding up samples from "plots" and "subplots" trying to be sure I understood where the data was coming from. I think the section would benefit from a paragraph in 2.2 that makes very clear: how many plots are there in each moraine? How far away are they from each other? (can this also be shown in Figure 1?) are there subplots in every plot or just the dye application plots? I realize this information is all included in the paper, but it's scattered throughout the methods so some piecing together was required for me to figure it out.

Response to Specific Comments Issue 2:
We agree with this comment and restructured this paragraph on the plot selection, soil sampling and subdivision of the plots during the irrigation experiment.
To clarify the study design further, we split Figure 2 into two Figures. The updated Figure 2 shows an illustration of the experimental design of the field campaign and contains the information about the plot selection and subdivision as well as the soil sampling scheme (see below).

[Figure]

**Figure 2: Illustration of the experimental design and soil sampling scheme at each moraine.**

We also made it clear in the caption of Figure 1 that we here show only one of the 3 plots per moraine. New caption text in Figure 1: *Location (left) and surface cover (right) of the four selected proglacial moraines of the Stein glacier. White circles show locations of one of the three brilliant blue experiment*

*plots per age class. Photo of location is provided by Google (n.d.). Photos of the 30, 160, and 10 000-year-old moraine were taken after the brilliant blue experiment (photos taken by F. Lustenberger).*

It is difficult to provide an overview of the plot locations at all moraines, since the plots were located several meters (10-100m) apart. But we will include this information in the text in Line 131:
*The distances between the three study plots at each moraine ranged from 10 to 100 m.*

Specific Comments Issue 3:
Heterogeneity. I think it'd be good to have more discussion about the heterogeneity in these moraines, then how that heterogeneity was addressed in the study design and how it affects the interpretation of the results. Would you expect these moraines to be pretty homogeneous? If not, how were the heterogeneities accounted for, and how likely is it that the results might be different if the sites were placed differently?

Response to Specific Comments Issue 3:
We agree with this comment and added at the end of the Discussion of the evolution of soil texture and structure the following paragraph from line 412 to 424 to address this issue:

*It is well known that soil properties are spatially heterogeneous [Bevington et al. (2016) Hu et al. (2008)]. As it was not possible to account for this variability with a large sample size, i.e. with a large number of experiments, we decided to take a different approach: Assuming that vegetation cover and subsurface flow paths are strongly linked, we took the variability in vegetation cover as a proxy and used it in an attempt to bracket this variability:  per moraine three locations that differ in their vegetation complexity (low, medium, high) were chosen for soil sampling and the dye tracer experiments. The analysis of the structural soil properties shows that there is a slight increase in spatial heterogeneity with age, especially in the top soil (increase in interquartile ranges for all properties in the top layer in Fig. 6), but occasionally also individual depths show a higher heterogeneity, irrespective of age.*
*The flow path analysis differentiated according to the vegetation complexity showed no systematic influence of the complexity level on the results. Heterogeneities within the individual experimental subplots were taken into account by averaging the volume density and surface area density across the five vertical profiles per subplot instead of relying on individual profiles. We therefore assume that the results of the flow path analysis are sufficiently representative to investigate their evolution across the chronosequence. .*

Specific Comments Issue 4:
Discussion structure. This discussion does a good job of putting the findings of the paper in context with previous work, but could you also add some information about how these changes are happening? Having processes tied to the changes would be really helpful for applying the findings here to other places. There is a bit of discussion about this with regards to vegetation and flowpaths, but not so much with the texture and structure. Additionally, in the first part of the discussion findings are sort of point-by-point related to previous literature. I wonder if the readability of the section might be improved by restructuring a bit to talk about how some of the changes in texture and structure happen together rather than breaking them all into separate paragraphs. There are a lot of findings here, and I realize that makes it kind of hard to present them concisely, so it's just a suggestion. But maybe similar processes are leading to the changes observed, and discussing those processes and the results may help.

Response to Specific Comments Issue 4:
We agree and restructured the discussion to improve readability. Instead of going through the soil physical characteristics one by one we are now discussing the different development stages jointly in all their characteristics.

We also added a paragraph on the processes affecting the soil texture and structure in Line 382 to 384:
*A high fraction of silt is very common for soils in mountain areas (Ellis, 1992). Physical weathering due to high fluctuations between day and night temperature and freezing cycles (Birse, 1980) leads to a reduction in grain size, without changing the particle mineralogy (Ellis, 1992).*

And in Line 356 to 362:
*After 160 years of soil development the porosity in the top layer increased and bulk density decreased. In general, these changes could be linked to changes in grain sizes, as the breakdown of particles leads to an increase in total pore space (porosity) and thus to a reduction in bulk density (Arvidsson, 1998). However, since changes in grain sizes were only marginal, the vegetation development, which includes an increase in root activities, litter accumulation, and biological activities in the root zone, is likely the main cause for changes in bulk density and porosity [Neris et al. (2012), Carey et al. (2007)].*

Technical Corrections
Page 6 Line 15 parameter should be plural
We agree and included the correction (see Line 172)
Page 6 Line 24 should "amount" be "number"?
We agree and included the correction (see Line 179)
Page 7 Line 6 I think the comma after "both" can be removed
We agree and included the correction (see Line 198)
Page 7 Lines 22 and 29 can you add a note explaining what you mean by disturbed and undisturbed? I assume the structure was preserved in the undisturbed sample…but they were both removed from the site, so they were definitely disturbed!
We included clarifications in section *2.4 Soil sampling and laboratory analysis* to point out that disturbed samples are samples taken without maintaining the natural soil structure and undisturbed samples were taken with sample rings to provide undisturbed cores which maintain the natural soil structure:

See Lines 213-214: *For grain size analysis, two disturbed bulk soil samples per depth were taken at 10, 30, and 50 cm depth at each plot.*

And Lines 224-225: *For the analysis of the structural parameters soil samples were taken with sample rings to provide undisturbed cores which preserve the natural soil structure.*

Page 16 Line 14 maybe "agree" would be better than "correspond"?
We agree and included the correction (see Line 379)
Page 18 Line 17 add "age" after "increasing moraine"
We agree and included the correction (see Line 458)
Page 18 Line 17 what is unstable flow?
We agree and included a further explanation in line 459 to 4608:
*Thus, we conclude that hydrophobicity of the organic top layer has a big impact on infiltration and the initiation of unstable flow. Unstable flow occurs when horizontal wetting fronts break into fingers or preferential flow paths during the downward movement (Hendrickx and Flury, 2001).*

Page 18 Line 27 maybe use a different word than "significantly" since it isn't a statistical
Comparison and Page 18 Line 27 see = saw

We agree and included the correction (see Line 473):
*At the oldest moraine, we saw a distinctly shallower infiltration depth.*
Page 20 Line 12 remove "also"
We agree and included the correction (see Line 506)
Page 20 Line 33 remove "already"
We agree and included the correction (see Line 530)

**Response to Reviewer 2, Nicholas Jarvis**

General Comments
This paper presents the results of dye tracing experiments showing how the mechanisms of preferential flow (PF) change with soil development in alpine moraines exposed by glacial retreat. This work culminates in a very nice schematic diagram (Figure 9) that summarizes and illustrates the main findings. The main strength of the paper is that the study of PF from this kind of pedological perspective is still really quite novel.
Nevertheless, the authors are not the first researchers to have taken this kind of pedological approach and it would strengthen the paper if some of this relevant earlier work could be mentioned in the Introduction. The authors could check out Quisenberry et al. (1993), Lin (2003), Cammaraat and Kooijman (2009) and Jarvis et al. (2102).
Another interesting and rather novel aspect of the paper is the demonstration of the importance of stones and rocks for generating and maintaining preferential flow. Maybe the authors could also cite Bogner et al. (2014), who demonstrated the same thing.
The paper is generally well written and presented and easy to read, although the language could be improved further by a native speaker. One minor concern is that the author's use of terminology related to PF is, at times, unnecessarily confusing. I have two other criticisms. First, the methods are not described in sufficient detail. Secondly, the authors could do a better job of discussing their results with respect to the fundamental processes causing the observed changes in flow patterns. These aspects are explained more fully in the following.

Response to General Comments
We thank the reviewer for spending his time to review and improve our manuscript. We will address all concerns and suggestions below.
We appreciate the references to further literature and included some of the references in line 82-84:
*This sort of hydropedological approach (Lin 2003) that links pedon (Quisenberry et al., 1993), landscape (Cammeraat and Kooijman, 2009) and hydrologic processes studies is likely to open up new insights into the preferential flow phenomenon (Jarvis et al. 2102).*

And in line 489 to 490 we included the sentence:
*The tendency of higher rock contents to increase the number of flow paths was also found by Bogner et al. (2014).*

Specific Comments Methods:
Methods: It was not clear to me what aspects of the vegetation cover were actually measured. For example, you use the term "mapped" on page 4 at line 15, but this is a rather vague. Do you have measurements of anything like above-ground biomass or was only species composition recorded? Please write this more explicitly. The description of vegetation complexity on page 5 at lines 2-4 is also not very helpful. Can you give a brief description here, maybe with an equation? The reader should not have to consult another paper (Musso et al.).

Response to Specific Comments:
We agree with this suggestion and added this explanation in lines 125 to 130:

*Three study plots were selected at each moraine, based on degree of vegetation complexity (low, medium and high complexity). Vegetation complexity is characterized by vegetation coverage, number of species and the plant functional diversity. The functional diversity is calculated based on specific leaf area, nitrogen content, leaf dry matter content, Raunkiaérs life form, seed mass, clonal growth organ, root type and growth form. The collection of*

*the required data and calculation of the vegetation complexity was done by the Geobotany Group of the University of Freiburg and is described in more detail in Maier et al. (2019).*

Please note that we moved this part from chapter 2.1 to chapter 2.2 Field experiments.

The description of the irrigation procedure on page 5 (lines 15-18) was quite difficult to Follow. It seems as if the irrigation pattern was different between the plots. Why was that? Perhaps a schematic figure might help to explain this.

Response to Specific Comments:
We agree with this comment and have clarified the procedure. We restructured the entire paragraph on plot selection and subdivision of the plots during the irrigation experiment to make it more clear that at each moraine three experimental plots were selected, which differ in vegetation complexity (low, medium, high). For the irrigation experiments each plot was further divided into three subplots for the application of three individual irrigation amounts.
We also split Figure 2 into two Figures and included an illustration of the dye tracer plot and the stepwise irrigation scheme into a separate Figure (now Figure 2) that shows the experimental set up of the field campaign in more detail:

[Figure]

**Figure 2: Illustration of the experimental design and soil sampling scheme at each moraine.**

We also updated the description of the irrigation procedure in lines 140 to 150 as follows:

*The tracer was applied with a hand-operated sprayer connected to a battery powered pump which guaranteed a constant pressure for a uniform flow rate of 60 l/h. For a time-efficient irrigation of the*

*three subplots with three irrigation amounts (20, 40, and 60 mm) and an intensity of 20 mm/h, the irrigation procedure was divided into three steps. In the first step all three subplots were irrigated simultaneously for 60 minutes in a sequence of 5 minutes irrigation and 5 minutes break. This provides an application of 20 mm to all three subplots. After finishing the first step the first subplot was covered to avoid any additional water input. In a second step, the other two subplots were simultaneously irrigated for additional 60 min in a sequence of 5 min irrigation and 10 min breaks. This provides an application of additional 20 mm to the two remaining subplots. In the last step only the third subplot was irrigated for 60 min in a sequence of 2 min irrigation and 10 min breaks while the other two plots remained covered providing an additional 20 mm to this last remaining subplot. After the end of tracer application, the entire plot was covered to avoid any disturbance by natural rainfall.*

The method to classify the flow patterns into different groupings is described only very briefly (on pages 6/7) and it was also difficult to follow. To complement the text (e.g. at lines 28-30 on page 6), could you give an equation or perhaps include a schematic diagram (or both)? This procedure is quite central to the paper, so it is important to explain this carefully.

Response to Specific Comments:
We agree with your suggestion and included a ternary diagram after Weiler (2001) (see below) that shows which flow type is assigned to which proportions of the 3 stained path width classes in terms of volume density in Line 191:

[Figure]

**Figure 4 Flow type classification based on the proportion of the three stained path width classes: ternary diagram after Weiler (2001).**

The description of the particle size analysis on page 7 makes no mention of the gravel/stone fraction (>2 mm). Did you measure the content of stones/gravel? I know this is very difficult in stony soils, so it is understandable if you didn't, but I think this should be stated.

Response to Specific Comments:
We added information about the measured gravel/stone fraction >2mm as a subfigure to the display of the soil texture results in Line 252:

[Figure]

**Figure 5. (a) Profile averaged grain size fractions for the four moraines. Fractions are percentages of the fine earth fraction (< 2 mm). (b) Profile averaged gravel content (> 2mm) calculated as the percentage of the entire sample weight. Each average is based on 18 samples.**

We also updated section 2.4 Soil sampling and laboratory analysis by the method used for the stone and gravel content measurements in line 220-223:
*Grain size fractions of particles < 2 mm were calculated as weight percentages of total weight of particles <2mm, thus excluding gravel and stones to avoid that single larger stones shift or dominate the distribution. The gravel and stone fraction was calculated separately as a weight percentage of the entire soil sample.*

The description of how bulk density and porosity were measured (page 7, lines 31-32) is quite vague. Can you describe more exactly (but still briefly) how you measured bulk density and (especially) porosity? It would be good to give some details, because the porosity values are extremely small in the young moraine and in the subsoils. I suppose this is because of the high stone content, but it could also be because air got trapped in the samples during saturation.

Response to Specific Comments:
We agree with this comment and included some further explanation on the methods used in line 229-233:

*The porosity was determined in the lab using the water saturation method. For this method sample weights were recorded at saturation and after drying at 105 °C. For saturation, the samples were placed in a small basin. The water level in the basin was increased step wise by 1 cm per day. When the water level reached the top of the soil sample and the sample was fully saturated, the bottom of the sample was sealed and the weight at saturation was measured. Bulk density was determined by relating the dry mass after drying at 105 °C to the sample volume.*

The authors do not report any measurements of soil organic matter content (SOM). This data should ideally be included in the paper, as the build-up of SOM over millennia due to the growth of vegetation seems to be a very important control on the observed changes in the flow patterns. If SOM was not measured, then I think it should be measured now and the results included in the paper (the analysis is quick and cheap).

Response to Specific Comments:
We have included the SOM data in the revised manuscript as an additional subfigure to the results of porosity and bulk density in Figure 6, Line 256:

[Figure]

**Figure 6. Evolution of soil porosity (a) bulk density (b) and loss on ignition (c) in 10, 30, and 50 cm depth.**

We further updated section 2.4 Soil sampling and laboratory analysis by the method used for the estimation of Loss on Ignition in line 233 to 236:

*The loss on ignition is a measure of the organic substance in the soil and describes the proportion of the organic substance that was oxidized during annealing for 24 hours at 550 °C. The loss on ignition was determined by drying sub-samples (4-6 g) for at least 24 hours at 105 °C and then at 550 °C. The ignition loss is then calculated by relating the weight loss after drying at 550 °C to the sample weight after drying at 105 °C.*

We also updated the results section 3.1 soil texture and structural parameters by the description of the results of loss on ignition in line 264 to 271:

*The loss on ignition, as a measure for the organic matter content, shows an increase throughout the first 10 millennia of soil development, which is most pronounced in the upper soil layer (see Fig. 6(c)). At the two youngest moraines the organic matter content is still very low (< 2 weight-%). At these two age classes the organic matter content is homogeneously distributed over the profile, with a slight tendency to higher values in the topsoil at the 160-year-old moraine. The 3000-year-old moraine shows a strong increase in organic matter content in the surface layer. At the oldest moraine the trend of increasing organic matter continues in all three depths. Here, the organic matter content in the topsoil makes up to two-thirds of the soil material. However, the organic matter content varies distinctly with a minimum of 6 weight-% and a maximum of 87 weight-%. In deeper depths, the organic content also increases compared to the 3 000 year old soil, but remains below 20 weight-%.*

Specific Comments Processes:
Processes: Can you explain (e.g. on page 16, lines 9-17) why the texture becomes finer with age? Is it due to weathering or is it deposition of fine materials by wind, or maybe both (or something else)? The cause(s) might be obvious to the authors, but perhaps they will not be to all readers.

Response to Specific Comments:
We agree and included some further clarification in line 382-384:

 A high fraction of silt is very common for soils in mountain areas (Ellis, 1992). Physical weathering due to high fluctuations between day and night temperature and freezing cycles (Birse, 1980) leads to a reduction in grain size, without changing the particle mineralogy (Ellis, 1992).

The process(es) operating to increase porosity and decrease bulk density should be explained better (e.g. on page 16, line 20, and lines 33-34). I presume that it is mostly related to a build-up of organic matter in the soil, which is supplied by the litter and roots of the increasingly dense vegetation cover and subsequently processed by soil micro-organisms and fauna, which ultimately results in a more open (aggregated) soil structure.

Response to Specific Comments:
We agree and included a sentence on that the linkage between the decrease in bulk density and vegetation development is caused by root activities, litter accumulation, and biological activities in the root zone in line 358-360:
*However, since changes in grain sizes were only marginal, the dense vegetation development, which includes an increase in root activities, litter accumulation, and biological activities in the root zone, is likely the main cause for changes in bulk density and porosity [Neris et al. (2012), Carey et al. (2007)].*

And in line 387-391:
*The continuous increase in porosity and reduction in bulk density can be attributed to the continuing change in soil texture on the one hand and on the other hand to the pronounced vegetation development. Especially the latter with the resulting accumulation of soil organic matter (see Fig. 6(c)) and the growth of an even denser root network that is now over 35 cm deep, is the main cause for the pronounced changes in the top soil.*

The authors associate the homogeneous flow patterns found in the young moraine with "gravity-driven" water flow (e.g. on page 15, line 2; page 18, line 2; page 20, line 21). This is rather misleading to my mind. Fundamentally, it must be the case that both gravity and capillarity were driving the infiltration process in all your experiments, because the soils were (presumably) initially quite dry. In fact, the authors do not really need to discuss whether gravity or capillarity dominated the flow patterns in the different moraines, but if they want to do so, then I think in reality, it is the opposite of what they write. Both macropore flow and finger flow are gravity-dominated processes, whereas a homogeneous flow pattern implies that capillarity was strong enough to prevent the development of any lateral non-equilibrium in soil water pressures. It is this lateral non-equilibrium in water pressures during flow that is a fundamental characteristic of PF.

Response to Specific Comments:
We removed the reference to gravity-driven flow. However, soils were generally quite wet during the entire field campaign as the total rainfall amount in this region is quite high and hence the soils never really dry out.

The coarse textured soil at the youngest moraine have a low water holding capacity and a high drainability. Therefore we updated our statements accordingly to the suggestions of the Reviewer in line 323-326:

*Using the information in the volume density profiles and the stained path widths to characterize flow types (Weiler, 2001) we found a trend from a rather homogeneous flow pattern with matrix flow in a fast draining coarse textured soil at the youngest moraine to a more heterogeneous flow pattern with a mix of heterogeneous matrix flow and finger flow at both medium age moraines (Fig. 9).*

And by removing line 442, and in line 513 to 515:

*The derived flow types also support our hypothesis that vertical subsurface flow path types change through the millennia. Flow types change from homogeneous matrix flow in a fast draining coarse textured soil to a heterogeneous matrix and finger flow over the first 100-3 000 years.*

Specific Comments Confusion over terminology:
Confusion over terminology: Considering the underlying physical mechanisms, there are three main types of PF (macropore flow, finger flow and heterogeneous flow) and this is indeed the basis of the classification system that the authors make use of in the paper.
However, the authors unnecessarily introduce some confusion at a couple of places in the paper by referring to another classification scheme, one that is not especially useful in my opinion:
i.) page 2 (lines 18/20): There is no good reason to distinguish crack flow from burrow flow (does burrow flow include flow in channels created by root decay?). These can all be lumped into macropore flow (as you do later). If you want to define some subgroups according to the origin of macropores, you should talk about flow in biopores (which includes both root and faunal channels) not burrow flow.

ii.) page 17, line 34: "In the clay layer, no significant macropores were identifiable, which is why it is assumed that the water is transported in cracks …..". Cracks are also macropores. You should replace the term macropores by biopores.

We agree with this comment and corrected our terminology accordingly in line 40-43:
*They defined four types of preferential flow: crack flow, burrow flow (created by soil fauna), finger flow, and lateral flow along layer interfaces, where flow in burrows and cracks is also often classified as macropore flow. We will in the following distinguish flow in macropores according to their origin as crack flow and biopore flow, where the latter includes channels by activities of roots and soil fauna.*

And in line 439-441:
*In the clay layer, no significant biopores were identifiable, which is why it is assumed that the water is transported in cracks or along material interfaces.*

Specific Comments Corrections:

1. The text at the end of the Introduction should be re-arranged. The hypotheses at lines 6-11 don't make much sense at the moment, because they are specific to the case of glacial moraines. It's not clear to the reader where these hypotheses come from. If you move this text to line 19 (after " … impacts water flow paths"), I think it will make more sense, especially if you add "… in glacial moraines in the Swiss Alps" after "… landscape evolution", and delete the last sentence in the first paragraph.

We agree and restructured the Introduction in accordingly in line 75-80.

2. Abstract, Line 1: you should delete "The presence or absence of …"

We agree and included the correction (see line 1)

3. page 3, line 1: add "volcanic" after "…younger"

We agree and included the correction (see line 57)

4. page 4, lines 15-16: delete "by the project partners … Germany"

We agree and included the correction (see line 112-113)

5. Page 7, line 10: maybe you could add "… and flow mechanisms" after "different properties"

We agree and included the correction (see line 202)

6. Page 8, line 1: add "…. moraines of differing …" after "four"

We agree and included the correction (see line 239)

7. Page 8, line 6: replace "the entire" by "all"

We agree and included the correction (see line 242)

8. Page 8, figure 3 caption: I presume that these results are % of the fine earth fraction (< 2mm). It would be good to state this here.

We agree and updated the caption of now Figure 5 accordingly:

(a) Profile averaged grain size fractions for the four moraines. Fractions are percentages of the fine earth fraction (< 2 mm). (b) Profile averaged gravel content (> 2mm) calculated as the percentage of the entire sample weight. Each average is based on 18 samples.

9. Page 10, line 5: I don't think you should talk about hillslopes as you haven't mentioned anything about site topography. You could just replace "hillslope" here by "moraine"

We agree and included the correction (see line 276)

10. Page 12, line 7: This is ambiguous, but I think you mean: "For all four moraines, the volume density is largest in the top half of the soil profile"

This was a misunderstanding due to ambiguous wording, we did want to state that the youngest moraine has a higher volume density in the top half of the soil profile than all the other moraines. We replaced "The volume density in the top half of the soil profile is the highest of all ages (Figure 5)" with the sentence "*The youngest moraine has a higher volume density of flow paths in the top half of the soil profile than all other moraines*" in Line 289.

11. Page 13, line 2: interpreting dye tracing patterns can be tricky, since you only get a snapshot in time of a dynamic process. In this particular case, I think it's possible that even if the staining was homogeneous, it doesn't necessarily mean that PF didn't occur. PF could have occurred from the soil surface, but the signs of this may have been obliterated by the later (slower) downward movement of a uniform wetting front in the soil matrix. I am not saying that this is what happened (I'm confident that your interpretation is correct), but I think you could recognize this possibility.

We agree and have added a sentence to account for this in line 157-160:
*Since dye tracer experiments only provide snapshots of flow patterns at 24 h after the irrigation, we cannot exclude the possibility that initial preferential flow paths were obliterated by a later downward movement of the infiltration front. However, as the probability for this special case is relatively low, we assume that these snapshots are a viable basis for the comparison of characteristic flow patterns along the moraine ages.*

12. Page 16, lines 26-27: I don't understand how the decrease in bulk density in the first 160 years can be related to a change of particle sizes, since this was marginal. It must be primarily due to the increase in SOM content.

We agree that this sentence is misleading and updated the statement in line 356-362:

*In general, these changes could be linked to changes in grainsizes, as the breakdown of particles leads to an increase in total pore space (porosity) and thus to a reduction in bulk density (Arvidsson, 1998). However, since changes in grain sizes were only marginal, the dense vegetation development, which includes an increase in root activities, litter accumulation, and biological activities in the root zone, is likely the main cause for changes in bulk density and porosity [Neris et al. (2012), Carey et al. (2007)].*

13. Page 16, lines 26-34: there is no need to have separate discussions for porosity and bulk density, because they are very closely linked (via the particle density). You could simplify and shorten the text between lines 18 and 34: you only need to write that the increase of porosity and decrease of bulk density was presumably a result of organic matter build-up in the soil due to the development of a denser vegetation cover.

We agree and restructured the Discussion section accordingly.

14. Page 17, lines 2-3: it should be briefly explained (with a supporting reference) how the change in texture could affect bulk density. Presumably the finer particles fill the spaces between the coarser particles? However, I think that the effects of texture on bulk density are usually considered to be relatively small. I think that the increase in SOM content (and associated biological activity in the soil) must be the main reason for the decrease in bulk density.

We agree and included some further clarification in line 356-362:

*In general, these changes could be linked to changes in particle sizes, as the breakdown of particles leads to an increase in total pore space (porosity) and thus to a reduction in bulk density (Arvidsson, 1998). However, since changes in grain sizes were only marginal, the dense vegetation development, which includes an increase in root activities, litter accumulation, and biological activities in the root zone, is likely the main cause for changes in bulk density and porosity [Neris et al. (2012), Carey et al. (2007)].*

15. Page 17, line 20: "texture" in this context is quite a vague term. Was it clay content? Please be more explicit.

We agree and changed the sentence in line 407 – 410 to:
*Saturated conductivity was found to be negatively correlated with the fraction of fine particles. The decrease in gravel content and the increase in silt seem to have an even a stronger effect on the saturated conductivity than the root network development (Maier et al., 2019).*

16. Page 17, line 30: the coarse nature of the material must be important too?
We agree and included the correction (see line 436)
17. Page 18, line 27: replace "lower" by "shallower"
We agree and included the correction (see line 472)
18. Page 18, line 32: should be: "… cover was removed to decrease …"
We agree and included the correction (see line 477)

[revised manuscript text omitted]